# BENO: BOUNDARY-EMBEDDED NEURAL OPERATORS FOR ELLIPTIC PDES

**Haixin Wang**[1,*]**, Jiaxin Li**[2,*]**, Anubhav Dwivedi**[3]**, Kentaro Hara**[3]**, Tailin Wu**[2,†]

[1]National Engineering Research Center for Software Engineering, Peking University,
[2]Department of Engineering, Westlake University,
[3]Department of Astronautics and Aeronautics, Stanford University
`wang.hx@stu.pku.edu.cn, lijiaxin@westlake.edu.cn,`
`{anubhavd,kenhara}@stanford.edu, wutailin@westlake.edu.cn`

## ABSTRACT

Elliptic partial differential equations (PDEs) are a major class of time-independent PDEs that play a key role in many scientific and engineering domains such as fluid dynamics, plasma physics, and solid mechanics. Recently, neural operators have emerged as a promising technique to solve elliptic PDEs more efficiently by directly mapping the input to solutions. However, existing networks typically cannot handle complex geometries and inhomogeneous boundary values present in the real world. Here we introduce **B**oundary-**E**mbedded **N**eural **O**perators (BENO), a novel neural operator architecture that embeds the complex geometries and inhomogeneous boundary values into the solving of elliptic PDEs. Inspired by classical Green's function, BENO consists of two branches of Graph Neural Networks (GNNs) for interior source term and boundary values, respectively. Furthermore, a Transformer encoder maps the global boundary geometry into a latent vector which influences each message passing layer of the GNNs. We test our model extensively in elliptic PDEs with various boundary conditions. We show that all existing baseline methods fail to learn the solution operator. In contrast, our model, endowed with boundary-embedded architecture, outperforms state-of-the-art neural operators and strong baselines by an average of 60.96%. Our source code can be found https://github.com/AI4Science-WestlakeU/beno.git.

## 1 INTRODUCTION

Partial differential equations (PDEs), which include elliptic, parabolic, and hyperbolic types, play a fundamental role in diverse fields across science and engineering. For all types of PDEs, but especially for elliptic PDEs, the treatment of boundary conditions plays an important role in the solutions. In particular, the Laplace and Poisson equations constitute prime examples of linear elliptic PDEs, which are used in a wide range of disciplines, including solid mechanics (Rivière, 2008), plasma physics (Chen, 2016), and fluid dynamics (Hirsch, 2007).

Recently, neural operators have emerged as a promising tool for solving elliptic PDEs by directly mapping input to solutions (Li et al., 2020b;c;a; Lötzsch et al., 2022). Lowering the computation efforts makes neural operators more attractive compared with classical approaches like finite element methods (FEM) (Quarteroni & Valli, 2008) and finite difference methods (FDM) (Dimov et al., 2015). However, existing neural operators have not essentially considered the influence of boundary conditions on solving elliptic PDEs. A distinctive feature of elliptic PDEs is their sensitivity to boundary conditions, which can heavily influence the behavior of solutions.

In fact, boundary conditions pose two major challenges for neural operators in terms of inhomogeneous boundary values and complex boundary geometry. **First**, inhomogeneous boundary conditions can cause severe fluctuations in the solution, and have a distinctive influence on the solution compared to the interior source terms. For example, as shown in Fig. 1, the inhomogeneous boundary

---

*Equal contribution. Work done as an intern at Westlake University. †Corresponding author.

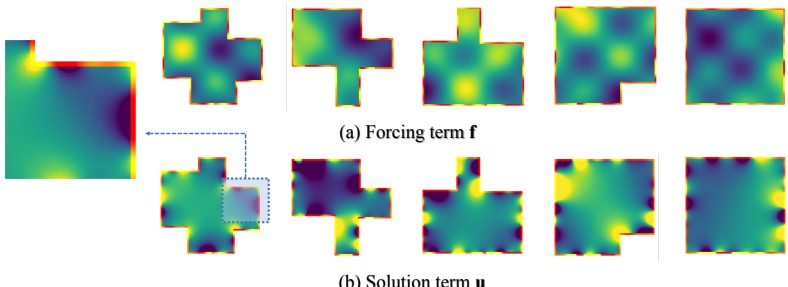

(a) Forcing term **f**

(b) Solution term **u**

Figure 1: Examples of different geometries for the elliptic PDEs: (a) forcing terms and (b) solutions. The nodes in red-orange color-map represent the complex, inhomogeneous boundary values. The redder the area, the higher the boundary value it represents, whereas the more orange the area, the lower the boundary value.

values cause high-frequency fluctuations in the solution especially near the boundary, which make it extremely hard to learn. **Second**, since elliptic PDEs are boundary value problems whose solution describes the steady-state of the system, any variation in the boundary geometry and values would influence the interior solution *globally* (Hirsch, 2007). The above challenges need to be properly addressed to develop a neural operator suitable for more general and realistic settings.

In this paper, we propose **B**oundary-**E**mbedded **N**eural **O**perators (BENO), a novel neural operator architecture to address the above two key challenges. Inspired by classical Green's function, BENO consists of two Graph Neural Networks (GNNs) that model the boundary influence and the interior source terms, respectively, addressing the first challenge. Moreover, to model the global influence of the boundary to the solution, we employ a Transformer (Vaswani et al., 2017) to encode the full boundary information to a latent vector and feed it to each message passing layer of the GNNs. This captures how the global geometry and values of the boundary influence the pairwise interaction between interior points, addressing the second challenge. As a whole, BENO provides a simple architecture for solving elliptic PDEs with complex boundary conditions, incorporating physics intuition into its boundary-embedded architecture. In Table 1, we provide a comparison between BENO and prior deep learning methods for elliptic PDE solving.

Table 1: Comparison of data-driven methods to time-independent elliptic PDE solving.

| Methods | 1. PDE-agnostic prediction on new initial condition | 2. Train/Test space grid independence | 3. Evaluation at unobserved spatial locations | 4. Free-form spatial domain for boundary shape | 5. Inhomogeneous boundary condition value |
|---|---|---|---|---|---|
| GKN (Li et al., 2020b) | ✓ | ✓ | ✓ | ✗ | ✗ |
| FNO (Li et al., 2020a) | ✓ | ✗ | ✓ | ✗ | ✗ |
| GNN-PDE (Lötzsch et al., 2022) | ✓ | ✓ | ✗ | ✓ | ✗ |
| MP-PDE (Brandstetter et al., 2022) | ✓ | ✗ | ✗ | ✗ | ✗ |
| **BENO (ours)** | ✓ | ✓ | ✓ | ✓ | ✓ |

To fully evaluate our model on inhomogeneous boundary value problems, we construct a novel dataset encompassing various boundary shapes, different boundary values, different types of boundary conditions, and varying resolutions. The experimental results demonstrate that our approach not only outperforms the existing state-of-the-art methods by about an average of 60.96% in solving elliptic PDEs problems but also exhibits excellent generalization capabilities in other scenarios. In contrast, all existing baselines fail to learn solution operators for the above challenging elliptic PDEs.

## 2 PROBLEM SETUP

In this work, we consider the solution of elliptic PDEs in a compact domain subject to inhomogeneous boundary conditions along the domain boundary. Let $u \in C^d(\mathbb{R})$ be a d-dimnesion-differentiable function of $N$ interior grid nodes over an open domain $\Omega$. Specifically, we consider the Poisson equation with Dirichlet (and Neumann in Appendix K) boundary conditions in a d-dimensional domain, and we consider $d = 2$ in the following experiments:

$$\begin{aligned} \nabla^2 u\left([x_1, x_2, \ldots, x_d]\right) &= f\left([x_1, x_2, \ldots, x_d]\right), & \forall \left([x_1, x_2, \ldots, x_d]\right) \in \Omega, \\ u\left([x_1, x_2, \ldots, x_d]\right) &= g\left([x_1, x_2, \ldots, x_d]\right), & \forall \left([x_1, x_2, \ldots, x_d]\right) \in \partial\Omega, \end{aligned}$$
(1)

where $f$ and $g$ are sufficiently smooth function defined on the domain $\Omega = \{(x_{1,i}, x_{2,i}, \ldots, x_{d,i})\}_{i=1}^{N}$, and boundary $\partial\Omega$, respectively. Eq. 1 is utilized in a range of applications in science and engineering to describe the equilibrium state, given by $f$ in the presence of time-independent boundary constraints specified by $g$. A distinctive feature of elliptic PDEs is their sensitivity to boundary values $g$ and shape $\partial\Omega$, which can heavily influence the behavior of their solutions. Appropriate boundary conditions must often be carefully prescribed to ensure well-posedness of elliptic boundary value problems.

## 3 METHOD

In this section, we detail our method BENO. We first motivate our method using Green's function, a classical approach to solving elliptic boundary value problems in Section 3.1. We then introduce our graph construction method in Section 3.2. Inspired by the Green's function, we introduce BENO's architecture in Section 3.3.

### 3.1 MOTIVATION

**How to facilitate boundary-interior interaction?** To design the boundary-embedded message passing neural network, we draw inspiration from the traditional Green's function (Stakgold & Holst, 2011) method which is based on a numerical solution. Take the Poisson equation with Dirichlet boundary conditions for example. Suppose the Green's function is $G : \Omega \times \Omega \to \mathbb{R}$, which is the solution of the corresponding equation as follows:

$$\begin{cases} \nabla^2 G = \delta(x - x_0)\delta(y - y_0) \\ G|_{\partial\Omega} = 0 \end{cases} \tag{2}$$

Based on the aforementioned equations and the detailed representation of the Green's function formula in the Appendix A, we can derive the solution in the following form:

$$u(x,y) = \iint_{\Omega} G(x,y,x_0,y_0)f(x_0,y_0)d\sigma_0 - \int_{\partial\Omega} g(x_0,y_0)\frac{\partial G(x,y,x_0,y_0)}{\partial n_0}dl_0 \tag{3}$$

Motivated by the two terms presented in Eq. 3, our objective is to approach boundary embedding by extending the Green's function. Following the mainstream work of utilizing GNNs as surrogate models (Pfaff et al., 2020; Eliasof et al., 2021; Lötzsch et al., 2022), we exploit the graph network simulator (Sanchez-Gonzalez et al., 2020) as the backbone to mimic the Green's function, and add the boundary embedding to the node update in the message passing. Besides, in order to decouple the learning of the boundary and interior, we adopt a dual-branch network structure, where one branch sets the boundary value $g$ to 0 to only learn the structural information of interior nodes, and the other branch sets the source term $f$ of interior nodes to 0 to only learn the structural information of the boundary. The Poisson equation solving can then be disentangled into two parts:

$$\begin{cases} \nabla^2 u(x,y) = f(x,y) \\ u(x,y) = g(x,y) \end{cases} \Rightarrow \underbrace{\begin{cases} \nabla^2 u(x,y) = f(x,y) \\ u(x,y) = 0 \end{cases}}_{\text{Branch 1}} + \underbrace{\begin{cases} \nabla^2 u(x,y) = 0 \\ u(x,y) = g(x,y) \end{cases}}_{\text{Branch 2}} \tag{4}$$

Therefore, our BENO will use a dual-branch design to build two different types of edges on the same graph separately. Branch 1 considers the effects of interior nodes and Branch 2 focuses solely on how to propagate the relationship between boundary values and interior nodes in the graph. Finally, we aggregate them together to obtain a more accurate solution under complex boundary conditions.

**How to embed boundary?** Since boundary conditions are crucially important for solving PDEs, how to better embed the boundary information into the neural network is key to our design. During a pilot study, we found that directly concatenating the interior node information with boundary information fails to solve for elliptic PDEs, and tends to cause severe over-fitting. Therefore, we propose to embed the boundary to represent its global information for further fusion. In recent years, Transformer (Vaswani et al., 2017) has been widely adopted due to its global receptive field. By leveraging its attention mechanism, the Transformer can effectively capture long-range dependencies and interactions within the boundary nodes. This is particularly advantageous when dealing with complex boundary conditions (i.e., irregular shape and inhomogeneous boundary values), as it allows for the modeling of complex relationships between boundary points and the interior solution.

## 3.2 Graph Construction

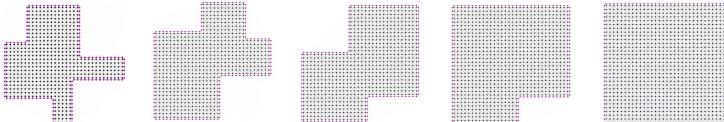

Figure 2: Visualization of the graph construction on our train/set samples from 5 different corner elliptic datasets. The interior nodes are in black and the boundary one in purple.

Before designing our method, it is an important step to construct graph $\mathcal{G} = \{(\mathcal{V}, \mathcal{E})\}$ with the finite discrete interior nodes as node set $\mathcal{V}$ on the PDE's solution domain $\Omega$. In traditional solution methods such as FEM, the solution domain is initially constructed by triangulating the mesh graph (Bern & Eppstein, 1995; Ho-Le, 1988), followed by the subsequent solving process. Therefore, the first step is to implement Delaunay triangulation (Lee & Schachter, 1980) to construct mesh graph with edge set $\mathcal{E}_{mesh}$, in which each cell consists of three edges. Then we proceed to construct the edge set $\mathcal{E}_{kn}$ by selecting the $K$-nearest nodes for each individual node. $K$ is the quantity of neighboring nodes that we deem as closely connected based on the Euclidean distance $\mathcal{D}_{ij}$ between node $i$ and $j$. The final edge set is $\mathcal{E} = \mathcal{E}_{mesh} \cup \mathcal{E}_{kn}$. Examples of graph construction are shown in Fig. 2.

## 3.3 Overall Architecture

In this section, we will introduce the detailed architecture of our proposed BENO, as shown in Figure 3. Our overall neural operator is divided into two branches, with each branch receiving different graph information and boundary data. However, the operator architecture remains the same with the *encoder, boundary-embedded message passing neural network* and *decoder*. Therefore, we will only focus on the common operator architecture.

### 3.3.1 Encoder & Decoder

**Encoder.** The encoder computes node and edge embeddings. For each node $i$, the node encoder $\epsilon^v$ maps the node coordinates $p_i = (x_i, y_i)$, forcing term $f_i$, and distances to boundary $dx_i, dy_i$ to node embedding vector $v_i = \epsilon^v([x_i, y_i, f_i, dx_i, dy_i]) \in R^D$ in a high-dimensional space. The same mapping is implemented on edge attributes with edge encoder $\epsilon^e$ for edge embedding vector $e_{ij}$. For both node and edge encoders $\epsilon$, we exploit a two-layer Multi-Layer Perceptron (MLP) (Murtagh, 1991) with Sigmoid Linear Unit (SiLU) activation (Elfwing et al., 2018).

**Decoder.** We use a two-layer MLP to map the features to solutions. Considering our dual-branch architecture, we will add the outputs from each decoder to obtain the final predicted solution $\hat{u}$.

### 3.3.2 Boundary-Embedded Message Passing Neural Network (BE-MPNN)

To address the inherent differences in physical properties between boundary and interior nodes, we opt not to directly merge these distinct sources of information into a single network representation. Instead, we first employ the Transformer to specifically embed the boundary nodes. Then, the obtained boundary information is incorporated into the graph message passing processor. We will provide detailed explanations for these two components separately.

**Embedding Boundary with Transformer.** With the boundary node coordinates $p^{\mathcal{B}} = (x^{\mathcal{B}}, y^{\mathcal{B}})$, the boundary value $g$, and the distance to the geometric center of solution domain $dc$ as input features, we first utilize the position embedding to include relative position relationship for initial representation $H_0^{\mathcal{B}}$, followed by a Transformer encoder with $L$ layers to embed the boundary information $H^{\mathcal{B}}$. The resulting boundary features, denoted as $\mathcal{B}$, are obtained by applying global average pooling (Lin et al., 2013) to the encoder outputs $H^{\mathcal{B}}$.

Each self-attention layer applies multi-head self-attention and feed-forward neural networks to the input. The output of the $i$-th self-attention layer is denoted as $H_i^{\mathcal{B}}$. The self-attention mechanism calculates the attention weights $A_i$ as follows:

$$A_i = \text{Softmax}\left( \frac{Q_i H_i^{\mathcal{B}}(K_i H_i^{\mathcal{B}})^T}{\sqrt{d_k}} \right) \tag{5}$$

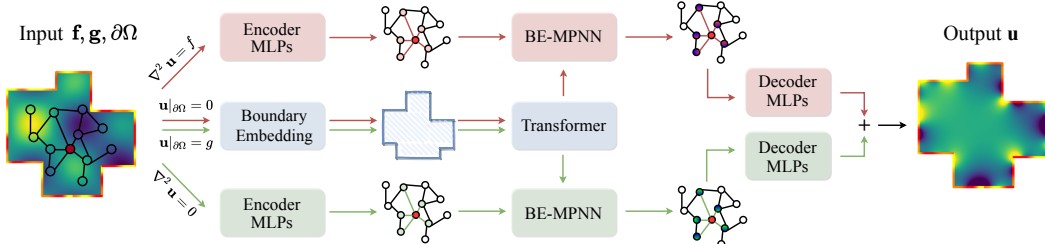

Figure 3: Overall architecture of our proposed BENO. The **pink branch** corresponds to the first term in Eq. 4, and the **green branch** corresponds to the second term. As the backbone of boundary embedding, **Transformer** provides boundary information as a supplement for BE-MPNN, thereby enabling better prediction under complex boundary geometry and inhomogeneous boundary values.

where $Q_i$, $K_i$, and $V_i$ are linear projections of $H_{i-1}^{\mathcal{B}}$ with learnable weight matrices, and $d_k$ is the dimension of the key vectors. The attention output is computed as:

$$H_{i+1}^{\mathcal{B}} = \text{LayerNorm}\left(A_i V_i \left(H_i^{\mathcal{B}}\right) + H_i^{\mathcal{B}}\right) \tag{6}$$

where LayerNorm denotes layer normalization, which helps to mitigate the problem of internal covariate shift. After passing through the $L$ self-attention layers, the output $H^{\mathcal{B}}$ is subject to global average pooling to obtain the boundary features: $\mathcal{B} = \text{AvgPool}(H^{\mathcal{B}})$.

**Boundary-Embedded Message Passing Processor.** The processor computes $T$ steps of message passing, with an intermediate graph representation $\mathcal{G}^1, \cdots, \mathcal{G}^T$ and boundary representation $\mathcal{B}^1, \cdots, \mathcal{B}^T$. The specific passing message $m_{ij}^t$ in step $t$ in our processor is formed by:

$$m_{ij}^t = \text{MLPs}\left(v_i^t, v_j^t, e_{ij}^t, p_i - p_j\right) \tag{7}$$

where $m_{ij}^{t+1}$ represents the message sent from node $j$ to $i$. $p_i - p_j$ is the relative position which can enhance the equivariance by justifying the symmetry of the PDEs.

Then we update the node feature $v_i^t$ and edge feature $e_{ij}^t$ as follows:

$$v_i^{t+1} = \text{MLPs}\left(v_i^t, \mathcal{B}^t, \sum_{j \in \mathcal{N}(i)} m_{ij}^t\right), \tag{8}$$

$$e_{ij}^{t+1} = \text{MLPs}\left(e_{ij}^t, m_{ij}^t\right) \tag{9}$$

Here, boundary information is embedded into the message passing. $\mathcal{N}(i)$ represents the gathering of all the neighbors of node $i$.

**Learning objective.** Given the ground truth solution $u$ and the predicted solution $\hat{u}$, we minimize the mean squared error (MSE) of the predicted solution on $\Omega$.

## 4 EXPERIMENTS

We aim to answer the following questions: (1) Compared with existing baselines, can BENO learn the solution operator for elliptic PDEs with complex geometry and inhomogeneous boundary values? (2) Can BENO *generalize* to out-of-distribution boundary geometries and boundary values, and different grid resolutions? (3) Are all components of BENO essential for its performance? We first introduce experiment setup in Sec. 4.1, then answer the above three questions in the following three sections.

### 4.1 EXPERIMENT SETUP

**Datasets.** For elliptic PDEs simulations, we construct five different datasets with inhomogeneous boundary values, including 4/3/2/1-corner squares and squares without corners. Each dataset consists of 1000 samples with randomly initialized boundary shapes and values, with 900 samples used for

Table 2: Performances of our proposed BENO and the compared baselines, which are trained on 900 4-corners samples and tested on 5 datasets under relative L2 norm and MAE separately. The unit of the MAE metric is $1 \times 10^{-3}$. Bold fonts indicate the best performance.

| Test set | Train on 4-Corners dataset | | | | | | | | | |
|---|---|---|---|---|---|---|---|---|---|---|
| | 4-Corners | | 3-Corners | | 2-Corners | | 1-Corner | | No-Corner | |
| Metric | L2 | MAE | L2 | MAE | L2 | MAE | L2 | MAE | L2 | MAE |
| GKN | 1.1146± 0.3936 | 3.6497± 1.1874 | 1.0692± 0.2034 | 3.7059± 0.9543 | 1.0673± 0.1393 | 3.6822± 0.9819 | 1.1063± 0.1905 | 3.4898± 0.9469 | 1.0728± 0.2074 | 3.9551± 0.9791 |
| FNO | 1.0947± 0.3265 | 2.2707± 0.3361 | 1.0742± 0.3418 | 2.1657± 0.3976 | 1.0672± 0.3736 | 2.2617± 0.2449 | 1.0921± 0.2935 | 2.3922± 0.3526 | 1.0762± 0.4420 | 2.2281± 0.4192 |
| GNN-PDE | 1.0026± 0.0093 | 3.1410± 0.8751 | 1.0009± 0.0101 | 3.2812± 0.8839 | 1.0015± 0.0099 | 3.3557± 0.8521 | 1.0002± 0.0153 | 3.1421± 0.8685 | 1.0011± 0.0152 | 3.7561± 1.0274 |
| MP-PDE | 1.0007± 0.0677 | 3.1018± 0.8431 | 1.0003± 0.0841 | 3.2464± 0.8049 | 0.9919± 0.0699 | 3.2765± 0.8632 | 0.9829± 0.07199 | 3.0163± 0.8272 | 0.9882± 0.0683 | 3.6522± 0.8961 |
| **BENO (ours)** | **0.3523± 0.1245** | **0.9650± 0.3131** | **0.4308± 0.1994** | **1.2206± 0.4978** | **0.4910± 0.1888** | **1.4388± 0.5227** | **0.5416± 0.2133** | **1.4529± 0.4626** | **0.5542± 0.1952** | **1.7481± 0.5394** |

training and validation, and 100 samples for testing. Each sample covers a grid of 32×32 nodes and 128 boundary nodes. To further assess model performance, higher-resolution versions of each data sample, such as 64×64, are also provided. Details on data generation are provided in Appendix C.

**Baselines.** We adopt two of the most mainstream series of neural PDE solvers as baselines, one is graph-based, including **GKN** (Li et al., 2020b), **GNN-PDE** (Lötzsch et al., 2022), and **MP-PDE** (Brandstetter et al., 2022); the other is operator-based, including **FNO** (Li et al., 2020a). For fair comparison and adaption to irregular boundary shapes in our datasets, all of the baselines are re-implemented with the same input as ours, including all the interior and boundary node features. Please refer to Appendix E for re-implementation details.

**Implementation Details.** All experiments are based on PyTorch (Paszke et al., 2019) and PyTorch-Geometric (Fey & Lenssen, 2019) on 2× NVIDIA A100 GPUs (80G). Following (Brandstetter et al., 2022), we also apply graph message passing neural network as our backbone for all the datasets. We use Adam (Kingma & Ba, 2014) optimizer with a weight decay of $5 \times 10^{-4}$ and a learning rate of $5 \times 10^{-5}$ obtained from grid search for all experiments. The relative L2 error measures the difference between the predicted and the ground truth values, normalized by the magnitude of the ground truth. MAE measures the average absolute difference between the predicted values and the ground truth values. Please refer to Appendix D for more implementation details.

## 4.2 MAIN EXPERIMENTAL RESULTS

We first test whether our BENO has a strong capability to solve elliptic PDEs with varying shapes. Table 2 and 3 summarize the results for the shape generalization task (more in Appendix H).

From the results, we see that recent neural PDE solving methods (i.e., MP-PDE) overall *fail* to solve elliptic PDEs with inhomogeneous boundary values, not to mention generalizing to datasets with different boundary shapes. This precisely indicates that existing neural solvers are insufficient for solving this type of boundary value problems.

In contrast, from Table 2, we see that our proposed BENO trained only on 4-Corners dataset consistently achieves a significant improvement and strong generalization capability over the previous methods by a large margin. More precisely, the improvements of BENO over the best baseline are 55.17%, 52.18%, 52.43%, 47.38%, and 52.94% in terms of relative L2 norm when testing on 4/3/2/1/No-Corner dataset respectively. We attribute the remarkable performance to two factors: (i) BENO comprehensively leverages boundary information, and fuses them with the interior graph message for solving. (ii) BENO integrates dual-branch architecture to fully learn boundary and interior in a decoupled way and thus improves generalized solving performance.

Similarly, from Table 3, we see that among mixed corner training results, BENO always achieves the best performance among various compared baselines when varying the test sets, which validates the consistent superiority of our BENO with respect to different boundary shapes.

Table 3: Performances of our proposed BENO and the compared baselines, which are trained on 900 mixed samples (180 samples each from 5 datasets) and tested on 5 datasets under relative L2 error and MAE separately. The unit of the MAE metric is $1 \times 10^{-3}$.

| Test set | \multicolumn{2}{c}{Train on mixed dataset} | | | | | | | | |
|---|---|---|---|---|---|---|---|---|---|---|
| Test set | 4-Corners | | 3-Corners | | 2-Corners | | 1-Corner | | No-Corner | |
| Metric | L2 | MAE | L2 | MAE | L2 | MAE | L2 | MAE | L2 | MAE |
| GKN | 1.0588± 0.1713 | 3.5051± 0.9401 | 1.0651± 0.1562 | 3.7061± 0.8563 | 1.0386± 0.1271 | 3.6043± 0.9392 | 1.0734± 0.1621 | 3.4048± 0.9519 | 1.0423± 0.2102 | 3.901± 0.9287 |
| FNO | 1.0834± 0.0462 | 4.6401± 0.5327 | 1.0937± 0.0625 | 4.6092± 0.6713 | 1.0672± 0.0376 | 4.5267± 0.5581 | 1.0735± 0.0528 | 4.5027± 0.5371 | 1.0713± 0.0489 | 4.5783± 0.5565 |
| GNN-PDE | 1.0009± 0.0036 | 3.1311± 0.8664 | 1.0003± 0.0039 | 3.2781± 0.8858 | 1.0005± 0.0038 | 3.3518± 0.8520 | 0.9999± 0.0042 | 3.1422± 0.8609 | 1.0002± 0.0041 | 3.7528± 1.0284 |
| MP-PDE | 1.0063± 0.0735 | 3.1238± 0.8502 | 1.0045± 0.0923 | 3.2537± 0.7867 | 0.9957± 0.0772 | 3.2864± 0.8607 | 0.9822± 0.0802 | 3.0177± 0.8363 | 0.9912± 0.0781 | 3.6658± 0.8949 |
| **BENO (ours)** | **0.4487± 0.1750** | **1.2150± 0.4213** | **0.4783± 0.1938** | **1.3509± 0.5432** | **0.4737± 0.1979** | **1.3516± 0.5374** | **0.5168± 0.1793** | **1.3728± 0.5148** | **0.4665± 0.2001** | **1.4213± 0.5262** |

Table 4: Performances of our BENO and the compared baselines, which are trained on 900 4-Corners samples and tested with zero boundary value samples. The unit of the MAE metric is $1 \times 10^{-3}$.

| Test set | \multicolumn{2}{c}{Train on 4-Corners dataset with homogeneous boundary value} | | | | | | | | |
|---|---|---|---|---|---|---|---|---|---|---|
| Test set | 4-Corners | | 3-Corners | | 2-Corners | | 1-Corner | | No-Corner | |
| Metric | L2 | MAE | L2 | MAE | L2 | MAE | L2 | MAE | L2 | MAE |
| GNN-PDE | 0.7092± 0.0584 | 0.1259± 0.0755 | 0.7390± 0.0483 | 0.2351± 0.1013 | 0.7491± 0.0485 | 0.3290± 0.1371 | 0.7593± 0.05269 | 0.4750± 0.1582 | 0.7801± 0.0371 | 0.6808± 0.1692 |
| MP-PDE | 0.2598± 0.1098 | 0.0459± 0.0359 | 0.3148± 0.0814 | 0.1066± 0.0618 | 0.3729± 0.0819 | 0.1778± 0.0969 | 0.4634± 0.0649 | 0.3049± 0.1182 | 0.5458± 0.0491 | 0.4924± 0.1310 |
| **BENO (ours)** | **0.0908± 0.07381** | **0.0142± 0.0131** | **0.1031± 0.0728** | **0.0288± 0.0189** | **0.1652± 0.1324** | **0.0583± 0.0362** | **0.1783± 0.1508** | **0.0862± 0.0456** | **0.2441± 0.1665** | **0.1622± 0.0798** |

Additionally, we plot the visualization of the best baseline and our proposed BENO trained on 4-Corners dataset in Figure 4. It can be clearly observed that the predicted solution of BENO is closed to the ground truth, while MP-PDE fails to learn any features of the solution. We observe similar behaviors for all other baselines.

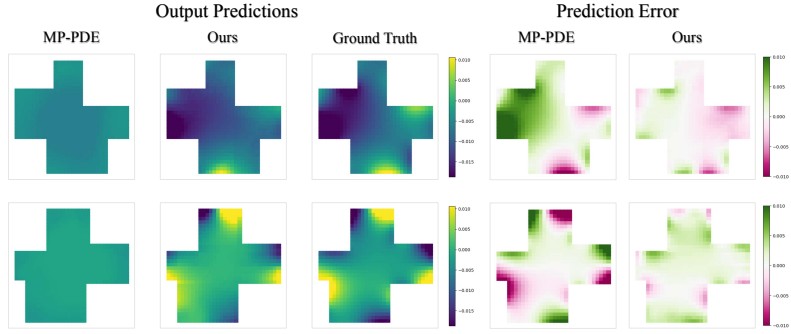

Figure 4: Visualization of two samples' prediction and prediction error from 4-Corners dataset. We render the solution $u$ of the baseline MP-PDE, our BENO and the ground truth in $\Omega$.

## 4.3 GENERALIZATION STUDY

### 4.3.1 RESULTS ON DIFFERENT BOUNDARY VALUES

To investigate the generalization ability on boundary value, we again train the models on 4-Corners dataset with inhomogeneous boundary value but utilize the test set with zero boundary value, which makes the boundary inhomogeneities totally different. Table 4 compares the best baseline and summarizes the results. From the results, we see that BENO has a significant advantage, successfully reducing the L2 norm to around 0.1. In addition, our method outperforms the best baseline by approximately 60.96% in terms of performance improvement. This not only demonstrates BENO's

Table 5: Performances of our BENO and the compared baselines, which are trained on 900 4-Corners $32 \times 32$ samples and tested with $64 \times 64$ samples. The unit of the MAE metric is $1 \times 10^{-3}$.

| | Train on 4-Corners dataset with $32 \times 32$ resolution | | | | | | | | | |
|---|---|---|---|---|---|---|---|---|---|---|
| Test set | 4-Corners($64 \times 64$) | | 3-Corners($64 \times 64$) | | 2-Corners($64 \times 64$) | | 1-Corner($64 \times 64$) | | No-Corner($64 \times 64$) | |
| Metric | L2 | MAE | L2 | MAE | L2 | MAE | L2 | MAE | L2 | MAE |
| MP-PDE | 0.6353± 0.1009 | 0.0596± 0.0418 | 0.7457± 0.0738 | 0.1138± 0.0533 | 0.7926± 0.0505 | 0.1565± 0.0596 | 0.8336± 0.04467 | 0.2445± 0.0915 | 0.8749± 0.0298 | 0.3991± 0.1045 |
| **BENO (ours)** | **0.4596±** **0.1094** | **0.0440±** **0.0349** | **0.5483±** **0.0987** | **0.0860±** **0.0466** | **0.6020±** **0.0842** | **0.1214±** **0.0537** | **0.6684±** **0.0794** | **0.1995±** **0.0851** | **0.7497±** **0.0653** | **0.3424±** **0.1000** |

Table 6: Ablation study of our BENO. The unit of the MAE metric is $1 \times 10^{-3}$.

| Test set | 4-Corners | | 3-Corners | | 2-Corners | | 1-Corner | | No-Corner | |
|---|---|---|---|---|---|---|---|---|---|---|
| Metric | L2 | MAE | L2 | MAE | L2 | MAE | L2 | MAE | L2 | MAE |
| **BENO** w. **M** | 1.0130± 0.0858 | 3.1436± 0.8667 | 1.0159± 0.0975 | 3.3041± 0.7906 | 0.9999± 0.0792 | 3.3007± 0.8504 | 1.0026± 0.0840 | 3.0842± 0.8202 | 0.9979± 0.0858 | 3.6832± 0.8970 |
| **BENO** w/o. **D** | 0.4058± 0.1374 | 1.1175± 0.3660 | 0.4850± 0.2230 | 1.3810± 0.6068 | 0.5273± 0.1750 | 1.5439± 0.4774 | 0.5795± 0.1981 | 1.5683± 0.4670 | 0.5835± 0.2232 | 1.8382± 0.5771 |
| **BENO** w. **E** | 0.4113± 0.1236 | 1.2020± 0.4048 | 0.4624± 0.2102 | 1.3569± 0.5453 | 0.5347± 0.1985 | 1.5990± 0.5604 | 0.5891± 0.2129 | 1.6222± 0.2016 | 0.5843± 0.2016 | 1.8790± 0.5952 |
| **BENO** w. **G** | 0.9037± 0.1104 | 2.6795± 0.5332 | 0.8807± 0.1298 | 2.6992± 0.6118 | 0.8928± 0.1208 | 2.8235± 0.5892 | 0.8849± 0.1462 | 2.561± 0.5085 | 0.8721± 0.1569 | 2.9851± 0.5591 |
| **BENO (ours)** | **0.3523±** **0.1245** | **0.9650±** **0.3131** | **0.4308±** **0.1994** | **1.2206±** **0.4978** | **0.4910±** **0.1888** | **1.4388±** **0.5227** | **0.5416±** **0.2133** | **1.4529±** **0.4626** | **0.5542±** **0.1952** | **1.7481±** **0.5394** |

strong generalization ability regarding boundary values but also provides solid experimental evidence for the successful application of our elliptic PDE solver.

### 4.3.2 DIFFERENT GRID RESOLUTIONS

Data-driven PDE solvers often face limitations in terms of the scale of the training data, making the ability to generalize to higher resolutions a crucial metric. Table 5 provides a summary of our performance in resolution generalization experiments. The model was trained on the 4-Corners homogeneous boundary value dataset with $32 \times 32$ resolution and tested with $64 \times 64$ samples not seen in training. The results demonstrate a significant advantage of our method over MP-PDE, with an improvement of approximately 22.46%. We attribute this advantage in generalization to two main factors. Firstly, it stems from the inherent capability of GNNs to process input graphs of various sizes. Secondly, it is due to our incorporation of relative positions as part of the network's edge features. Consequently, our approach can be deployed on different resolutions using the same setup.

### 4.4 ABLATION STUDY

To investigate the effectiveness of inner components in BENO, we study four variants of BENO. Table 6 shows the effectiveness of our BENO on ablation experiments, which is implemented based on 4-Corners dataset training. Firstly, **BENO** w. **M** replaces the BE-MPNN with a vanilla message passing neural network (Gilmer et al., 2017) and merely keeps the interior node feature. Secondly, **BENO** w/o. **D** removes the dual-branch structure of BENO and merely utilizes a single Encoder-BE-MPNN-Decoder procedure. Thirdly, **BENO** w. **E** adds the Transformer output for edge message passing. Finally, **BENO** w. **G** replaces the Transformer architecture with a vanilla graph convolution network (Kipf & Welling, 2016).

From the results we can draw conclusions as follows. Firstly, **BENO** w. **M** performs significantly worse than ours, which indicates the importance of fusing interior and boundary in BENO. Secondly, comparing the results of **BENO** w/o. **D** with ours we can conclude that decoupled learning of the interior and boundary proves to be effective. Thirdly, comparing the results of **BENO** w. **E** and ours, we can find that boundary information only helps in node-level message passing. In other words, it is not particularly suitable to directly inject the global information of the boundary into the edges. Finally, comparing results of **BENO** w. **G** with ours validates the design of Transformer for boundary embedding is crucial.

## 5 RELATED WORK

### 5.1 CLASSIC ELLIPTIC PDE SOLVERS

The classical numerical solution of elliptic PDEs approximates the domain $\Omega$ and its boundary $\partial\Omega$ in Eq. 1 using a finite number of non-overlapping partitions. The solution to Eq. 1 is then approximated over these partitions. A variety of strategies are available for computing this discrete solution. Popular approaches include finite volume method (FVM) (Hirsch, 2007), finite element method (FEM) (Hughes, 2012), and finite difference method (FDM) (LeVeque, 2007). In the present work we utilize the FVM to generate the dataset which can easily accommodate complex boundary shapes. This approach partitions the domains into cells, and the boundary is specified using cell interfaces. After numerically approximating the operator $\nabla^2$ over these cells, the numerical solution is obtained on the centers of the cells constituting our domain. Further details are provided in Appendix B.

### 5.2 GNN FOR PDE SOLVER

GNNs are initially applied in physics-based simulations on solids and fluids represented by particles (Sanchez-Gonzalez et al., 2018). Recently, an important advancement MeshGraphNets (Pfaff et al., 2020) emerge to learn mesh-based simulations. Subsequently, several variations have been proposed, including techniques for accelerating finer-level simulations by utilizing GNNs (Belbute-Peres et al., 2020; Yang & Hong, 2022), combining GNNs with Physics-Informed Neural Networks (PINNs) (Gao et al., 2022), solving inverse problems with GNNs and autodecoder-style priors (Zhao et al., 2022), and handling temporal distribution shift (Luo et al., 2023). However, the research focus on addressing boundary issues is limited. T-FEN (Lienen & Günnemann, 2022), FEONet (Lee et al., 2023), VQGraph Yang et al. (2024) and GNN-PDE (Lötzsch et al., 2022) are pioneering efforts in this regard, encompassing complex domains and various boundary shapes. Nevertheless, the boundary values are still set to zero, which does not account for the presence of inhomogeneous boundary values. This discrepancy is precisely the problem that we aim to address.

### 5.3 NEURAL OPERATOR AS PDE SOLVER

Neural operators map from initial/boundary conditions to solutions through supervised learning in a mesh-invariant manner. Prominent examples of neural operators include the Fourier neural operator (FNO) (Li et al., 2020a), graph neural operator (Li et al., 2020b), and DeepONet(Lu et al., 2019). Neural operators exhibit invariance to discretization, making them highly suitable for solving PDEs. Moreover, neural operators enable the learning of operator mappings between infinite-dimensional function spaces. Subsequently, further variations have been proposed, including techniques for solving arbitrary geometries PDEs with both the computation efficiency and the flexibility (Li et al., 2022), enabling deeper stacks of Fourier layers by independently applying transformations (Tran et al., 2021), utilizing Fourier layers as a replacement for spatial self-attention (Guibas et al., 2021), facilitating boundary condition satisfaction in neural operators by implementing structural modifications to the operator kernel (Saad et al., 2022) and incorporating symmetries in the physical domain using group theory (Helwig et al., 2023). (Gupta et al., 2021; 2022; Xiao et al., 2023) continuously improve the design of the operator by introducing novel methods for numerical computation.

## 6 CONCLUSION

In this work, we have proposed Boundary-Embedded Neural Operators (BENO), a neural operator architecture to address the challenges posed by inhomogeneous boundary conditions with complex boundary geometry in solving elliptic PDEs. Our approach BENO incorporates physics intuition through a boundary-embedded architecture, consisting of GNNs and a Transformer, to model the influence of boundary conditions on the solution. By constructing a diverse dataset with various boundary shapes, values, and resolutions, we have demonstrated the effectiveness of our approach in outperforming existing state-of-the-art methods by an average of 60.96% in solving elliptic PDE problems. Furthermore, our method BENO exhibits strong generalization capabilities across different scenarios. The development of BENO opens up new possibilities for efficiently and accurately solving elliptic PDEs with complex boundary conditions, making them more useful to various scientific and engineering fields.

## ACKNOWLEDGEMENT

We gratefully acknowledge the support of Westlake University Research Center for Industries of the Future, and Westlake University Center for High-performance Computing.

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

APPENDIX

## A  DERIVATION OF THE GREEN'S FUNCTION METHOD.

We first review the definition of the Green's function, which is $G : \Omega \times \Omega \to \mathbb{R}$, which is the solution of the corresponding equation as follows:

$$\begin{cases} \nabla^2 G = \delta(x - x_0)\delta(y - y_0) \\ G|_{\partial\Omega} = 0 \end{cases} \tag{10}$$

According to Green's identities,

$$\iint_\Omega (u\nabla^2 G)d\sigma = \int_{\partial\Omega} u\frac{\partial G}{\partial n}dl - \iint_\Omega (\nabla u \cdot \nabla G)d\sigma \tag{11}$$

Since $u$ and $G$ are arbitrary, we can change the position to obtain that,

$$\iint_\Omega (G\nabla^2 u)d\sigma = \int_{\partial\Omega} G\frac{\partial u}{\partial n}dl - \iint_\Omega (\nabla u \cdot \nabla G)d\sigma \tag{12}$$

Subtract Eq. 12 from Eq. 11, we have,

$$\iint_\Omega (u\nabla^2 G - G\nabla^2 u)d\sigma = \int_{\partial\Omega}\left(u\frac{\partial G}{\partial n} - G\frac{\partial u}{\partial n}\right)dl \tag{13}$$

Substitute Eq. 13 into Eq. 10, we can have that,

$$\begin{aligned} \int_{\partial\Omega}\left(u\frac{\partial G}{\partial n} - G\frac{\partial u}{\partial n}\right)dl &= \iint_\Omega (u \cdot \nabla^2 G - G \cdot \nabla^2 u)d\sigma \\ &= \iint_\Omega (-u\delta(x - x_0)\delta(y - y_0) - G\nabla^2 u)d\sigma \\ &= -u(x_0, y_0) - \iint_\Omega G\nabla^2 u d\sigma \\ &= -u(x_0, y_0) + \iint_\Omega Gf(x, y)d\sigma \end{aligned} \tag{14}$$

Namely, we have that,

$$\begin{aligned} u(x, y) &= \iint_\Omega G(x, y, x_0, y_0)f(x_0, y_0)d\sigma_0 \\ &+ \int_{\partial\Omega}\left[G(x, y, x_0, y_0)\frac{\partial u(x_0, y_0)}{\partial n_0} - u(x_0, y_0)\frac{\partial G(x, y, x_0, y_0)}{\partial n_0}\right]dl_0 \end{aligned} \tag{15}$$

When considering the Dirichlet boundary conditions, we can simplify the solution in the following form:

$$u(x, y) = \iint_\Omega G(x, y, x_0, y_0)f(x_0, y_0)d\sigma_0 - \int_{\partial\Omega} g(x_0, y_0)\frac{\partial G(x, y, x_0, y_0)}{\partial n_0}dl_0 \tag{16}$$

## B  NUMERICAL SOLUTION OF THE ELLIPTIC PDE

The strong solution to equation 1 can be expressed in terms of the Green's function (see Section 3.1 and Appendix A for discussion). However, obtaining a closed form expression using the Green's function is typically not possible, except for some limited canonical domain shapes. In the present paper, we obtain the solution to equation 1 in arbitrary two dimensional domains $\Omega$ using the finite volume method (Hirsch, 2007). This numerical approach relies on discretizing the domain $\Omega$ using *cells*. The surfaces of these cells at the boundary, which are called *cell interfaces*, are used to specify the given boundary condition. The solution of equation 1 is then numerically approximated over $N$ (e.g., for 32×32 cells, $N = 1024$) computational cells by solving,

$$\mathbf{P\hat{u}} = \mathbf{f}, \tag{17}$$

where $\mathbf{P} \in \mathbb{R}^{N \times N}$ is an $N \times N$ matrix which denotes a second-order discretization of the $\nabla^2$ operator incorporating the boundary conditions, $\hat{\mathbf{u}} \in \mathbb{R}^{N \times 1}$ is a vector of values at the cell centers, and $\mathbf{f} \in \mathbb{R}^{N \times 1}$ is a vector with values $f(\cdot, \cdot)$ at cell centers used to discretize the domain $\Omega$. The matrix $\mathbf{P}$ resulting from this approach is positive definite and diagonally dominant, making it convenient to solve Equation 17 with a matrix-free iterative approach such as the Gauss-Seidel method (Saad, 2003).

## C    DETAILS OF DATASETS

In this paper, we have established a comprehensive dataset for solving elliptic PDEs to facilitate various research endeavors. The elliptic PDEs solver is performed as follows. (1) A square domain is set with $N_c$ number of cells in both $x$ and $y$ directions (note $N = N_c^2$). The number of corners is set, however, the size of the corners is chosen randomly. (2) The source term $f(x, y)$ is assigned assuming a variety of basis functions, including sinusoidal, exponential, logarithmic, and polynomial distributions. (3) The values of the boundary conditions $g(x, y)$ are set using continuous periodic functions with a uniformly distributed wavelength $\in [1, 5]$. (4) The Gauss-Seidel method (Saad, 2003) is used to iteratively obtain the solution $u(x, y)$. Each Poisson run generates two files: one for the interior cells with discrete values of $x$, $y$, $f$, and $u$ and the other for the boundary interfaces with discrete values of $x$, $y$, and $g$. The simulations are performed on the Sherlock cluster at Stanford University.

## D    MORE IMPLEMENTATION DETAILS

Our normalization process is performed using the z-score method (Patro & Sahu, 2015), where the mean and standard deviation are calculated from the training set. This ensures that all features are normalized based on the mean and variance of the training data. We also apply the CosineAnnealing-WarmRestarts scheduler (Loshchilov & Hutter, 2016) during the training. Each experiment is trained for 1000 epochs, and validation is performed after each epoch. For the final evaluation, we select the model parameters from the epoch with the lowest validation loss. Consistency is maintained across all experiments by utilizing the same random seed.

All our experiments are evaluated on relative L2 error, abbreviated as L2, and mean absolute error (MAE), which are two commonly used metrics for evaluating the performance of models or algorithms. The relative L2 error, also known as the normalized L2 error, measures the difference between the predicted values and the ground truth values, normalized by the magnitude of the ground truth values. It is typically calculated as the L2 norm of the difference between the predicted and ground truth values, divided by the L2 norm of the ground truth values. On the other hand, MAE measures the average absolute difference between the predicted values and the ground truth values. It is calculated by taking the mean of the absolute differences between each predicted value and its corresponding ground truth value.

## E    DETAILS OF BASELINES

Our proposed BENO is compared with a range of competing baselines as follows:

- **GKN** (Li et al., 2020b) develops an approximation method for mapping in infinite-dimensional spaces by combining non-linear activation functions with a set of integral operators. The integration of kernels is achieved through message passing on graph networks. For fair comparison, we re-implement it by adding the boundary nodes to the graph. To better distinguish between nodes belonging to the interior and those belonging to the boundary, we have also added an additional column of one-hot encoding to the nodes for differentiation.

- **FNO** (Li et al., 2020a) introduces a novel approach that directly learns the mapping from functional parametric dependencies to the solution. The method implements a series of layers computing global convolution operators with the fast Fourier transform (FFT) followed by mixing weights in the frequency domain and inverse Fourier transform, enabling an architecture that is both expressive and computationally efficient. For fair comparison, we re-implement it by fixing the value of out-domain nodes with a large number, and then implement the global operation.

- **GNN-PDE** (Lötzsch et al., 2022) represents the pioneering effort in training neural networks on simulated data generated by a finite element solver, encompassing various boundary shapes. It evaluates the generalization capability of the trained operator across previously unobserved scenarios by designing a versatile solution operator using spectral graph convolutions. For fair comparison, we re-implement it by adding the boundary nodes to the graph. To better distinguish between nodes belonging to the interior and those belonging to the boundary, we have also added an additional column of one-hot encoding to the nodes for differentiation.

- **MP-PDE** (Brandstetter et al., 2022) presents a groundbreaking solver that utilizes neural message passing for all its components. This approach replaces traditionally heuristic-designed elements in the computation graph with neural function approximators that are optimized through backpropagation. For fair comparison, we re-implement it by adding the boundary nodes to the graph. To better distinguish between nodes belonging to the interior and those belonging to the boundary, we have also added an additional column of one-hot encoding to the nodes for differentiation.

## F    DIFFERENCES WITH OTHER NEURAL OPERATORS

In this section, we compare our method, BENO, with existing approaches in terms of several key aspects according to Table 1.

- PDE-agnostic prediction on new initial conditions: GKN, FNO, GNN-PDE, MP-PDE, and BENO are all capable of predicting new initial conditions.

- Train/Test space grid independence: GKN, GNN-PDE, and BENO exhibit independence between the training and testing spaces, while FNO and MP-PDE lack this independence.

- Evaluation at unobserved spatial locations: GKN, FNO, and BENO are capable of evaluating the PDE at locations that are not observed during training, while GNN-PDE and MP-PDE do not possess this capability.

- Free-form spatial domain for boundary shape: Only GNN-PDE and BENO are capable of dealing with arbitrary boundary shapes, while GKN and MP-PDE are limited in this aspect.

- Inhomogeneous boundary condition value: Only our method, BENO, has the ability to handle inhomogeneous boundary conditions, while GKN, FNO, GNN-PDE, and MP-PDE are unable to handle them.

In summary, compared to the existing methods, our method, BENO, possesses several distinct advantages. It can predict new initial conditions regardless of the specific PDE, maintains grid independence between training and testing spaces, allows evaluation at unobserved spatial locations, handles free-form spatial domains for boundary shapes, and accommodates inhomogeneous boundary condition values. These capabilities make BENO a versatile and powerful approach for solving time-independent elliptic PDEs.

## G    ALGORITHM

The whole learning algorithm of BENO is summarized in Algorithm 1.

## H    MORE EXPERIMENTAL RESULTS

### H.1    SENSITIVITY ANALYSIS

In this section, we discuss the process of determining the optimal values for the number of MLP layers ($M$) and the number of Transformer layers ($L$) using grid search, a systematic approach for hyper-parameter tuning.

Grid search involves defining a parameter grid consisting of different combinations of $M$ and $L$ values. We specified $M$ in the range of [2, 3, 4] and $L$ in the range of [1, 2, 3] to explore a diverse set of configurations. We build multiple models, each with a different combination of $M$ and $L$ values, and train them on 4-Corners training dataset. The models are then evaluated using appropriate

---

**Algorithm 1** Learning Algorithm of the proposed BENO

---

**Input:** The forcing term $f$, the inhomogeneous boundary condition $g$ on $\partial\Omega$ .
**Output**: The solution prediction $\hat{u}$ of the elliptic PDEs.

 1: Construct the graph $\mathcal{G} = \{(\mathcal{V}, \mathcal{E})\}$ following Section 3.2;
 2: Initialize the parameters in our model;
    # Training procedure
 3: **while** not convergence **do**
 4:    **for** each training input **do**
 5:        Set the boundary value of one branch to zero following Eq. 16;
 6:        Set the source term of interior in the other branch to zero;
 7:        Feed the node/edge attributes to encoder following Section 3.3.1.;
 8:        Feed the boundary to the Transformer for boundary features $\mathcal{B}$;
 9:        Add $\mathcal{B}$ to the message passing processor following Eq. 8;
10:        Feed output features into a decoder to get the predictions $\hat{u}$;
11:        Calculate the loss using MSE;
12:        Update the parameters in BENO using back propagation;
13:    **end for**
14: **end while**

---

evaluation metrics on a separate validation set. The evaluation results allowed us to compare the performance of models across different parameter combinations.

After evaluating the models, we select the combination of $M$ and $L$ that yield the best performance according to our chosen evaluation metric. This combination became our final choice for $M$ and $L$, representing the optimal configuration for our model. To ensure the reliability of our chosen parameters, we validate them on an independent validation set. This step confirmed that the model's performance remained consistent and reliable.

The grid search process provided a systematic and effective approach to determine the optimal values for $M = 3$ and $L = 1$, allowing us to fine-tune our model and achieve improved performance.

### H.2 MORE EXPERIMENTAL RESULTS

We have successfully validated our method's performance on the 4-Corners and mixed corners datasets during training and testing on other shape datasets, yielding favorable results. In this section, we will further supplement the evaluation by training on the No Corner dataset and testing on other shape datasets. Since the No Corner dataset does not include any corner scenarios, the remaining datasets present completely unseen scenarios for it, thereby providing a stronger test of the model's generalization performance.

Table 7 summarizes the results of training on 900 No-Corner samples and tested on all datasets. We can infer similar conclusions to those in the experimental section above. Our BENO performs well in learning on No-Corner cases, yielding more accurate solutions. Additionally, our method demonstrates stronger generalization ability, as it can obtain good solutions even in cases where corners of any shape have not been encountered.

### H.3 CONVERGENCE ANALYSIS

We draw the training curve of the train L2 norm and test L2 norm of three models trained on the 4-Corners dataset with inhomogeneous boundary value in Figure 5. It is obviously that although two baselines also contains the boundary information, they fail to learn the elliptic PDEs with non-decreasing convergence curves. However, our proposed BENO is capable of successfully learning complex boundary conditions with the use of the CosineAnnealingWarmRestarts scheduler, converges to a satisfactory result.

Table 7: Performances of our proposed BENO and the compared baselines, which are trained on 900 No-Corner samples and tested on 5 datasets under relative L2 Norm and MAE separately. The unit of the MAE metric is $1 \times 10^{-3}$.

| Test set | Train on No-Corner dataset | | | | | | | | | |
|---|---|---|---|---|---|---|---|---|---|---|
| | 4-Corners | | 3-Corners | | 2-Corners | | 1-Corner | | No-Corner | |
| Metric | L2 | MAE | L2 | MAE | L2 | MAE | L2 | MAE | L2 | MAE |
| GKN | 1.0147± 0.1128 | 3.3790± 0.8922 | 1.0179± 0.1212 | 3.5419± 0.8643 | 1.0047± 0.1166 | 3.4530± 0.9514 | 1.0072± 0.1098 | 3.2295± 0.8520 | 1.0028± 0.1060 | 3.6899± 0.8987 |
| FNO | 0.9714± 0.0128 | 3.3210± 0.6546 | 0.9745± 0.0175 | 3.3187± 0.6639 | 0.9733± 0.0137 | 3.3319± 0.6298 | 0.9789± 0.0210 | 3.3511± 0.6109 | 0.9755± 0.0121 | 3.3427± 0.6981 |
| GNN-PDE | 0.9988± 0.0051 | 3.1182± 0.8543 | 0.9997± 0.0054 | 3.2748± 0.8902 | 0.9994± 0.0054 | 3.3475± 0.8533 | 1.0002± 0.0056 | 3.1447± 0.8559 | 0.9998± 0.0056 | 3.7518± 1.0314 |
| MP-PDE | 1.0029± 0.0808 | 3.1005± 0.8158 | 1.0049± 0.0891 | 3.2488± 0.7941 | 0.9986± 0.0822 | 3.2902± 0.8651 | 0.9855± 0.0769 | 3.0356± 0.8133 | 0.9917± 0.07670 | 3.6648± 0.8949 |
| **BENO (ours)** | **0.6870± 0.2038** | **1.8830± 0.6083** | **0.6036± 0.1940** | **1.7293± 0.5844** | **0.5760± 0.1998** | **1.6703± 0.6605** | **0.6192± 0.2259** | **1.6749± 0.5773** | **0.4093± 0.1873** | **1.2505± 0.5752** |

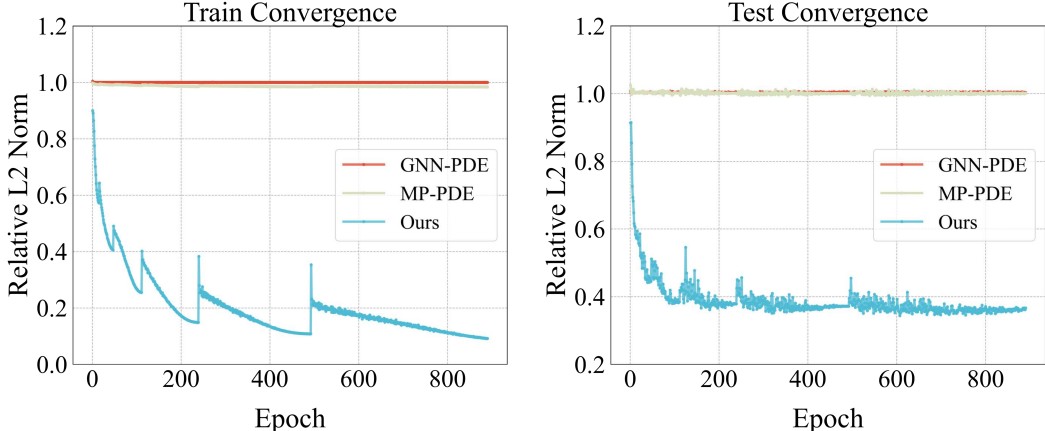

Figure 5: Visualization of the convergence curve of our BENO and two baselines.

# I  LIMITATIONS & BROADER IMPACTS

**Limitations.** Although this paper primarily focuses on Dirichlet boundary conditions, it is essential to acknowledge that there are other types of boundary treatments, including Neumann and Robin boundary conditions. While the framework presented in this study may not directly address these alternative boundary conditions, it still retains its usefulness. Future research should explore the extension of the developed framework to incorporate these different boundary treatments, allowing for a more comprehensive and versatile solution for a broader range of practical problems.

**Broader Impact.** The development of a fast, efficient, and accurate neural network for solving PDEs holds significant potential for numerous physics and engineering disciplines. The impact of such advancements cannot be understated. By providing a more streamlined and computationally efficient approach, this research can revolutionize fields such as computational fluid dynamics, solid mechanics, electromagnetics, and many others. The ability to solve PDEs more efficiently opens up new possibilities for modeling and simulating complex physical systems, leading to improved designs, optimizations, and decision-making processes. The resulting advancements can have far-reaching implications, including the development of more efficient and sustainable technologies, enhanced understanding of natural phenomena, and improved safety and reliability in engineering applications. It is crucial to continue exploring and refining these neural network-based approaches to maximize their potential impact across a wide range of scientific and engineering disciplines.

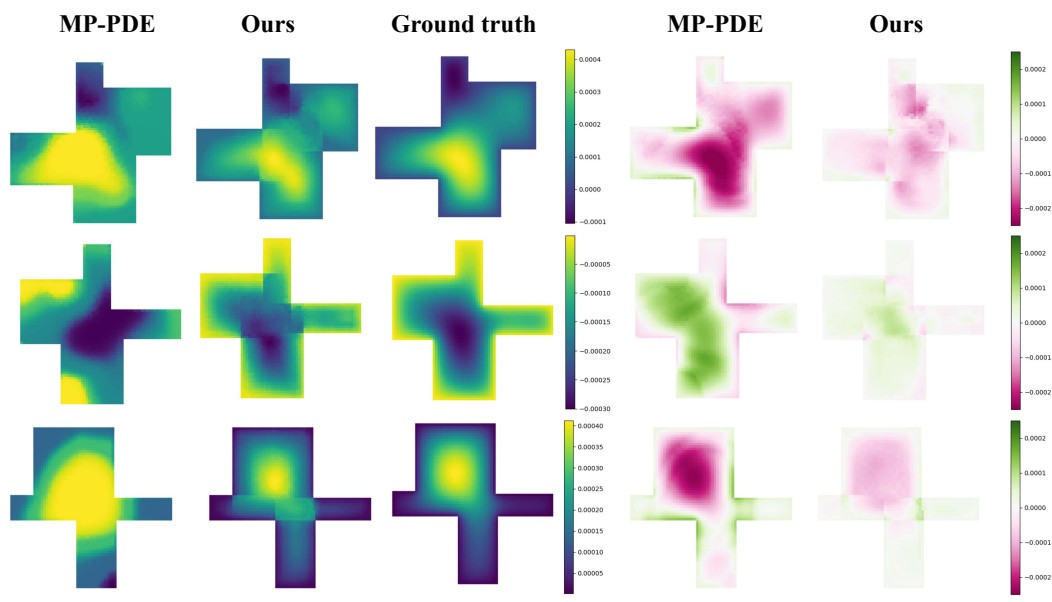

Figure 6: Visualization of prediction and prediction error on $64 \times 64$ grid resolution.

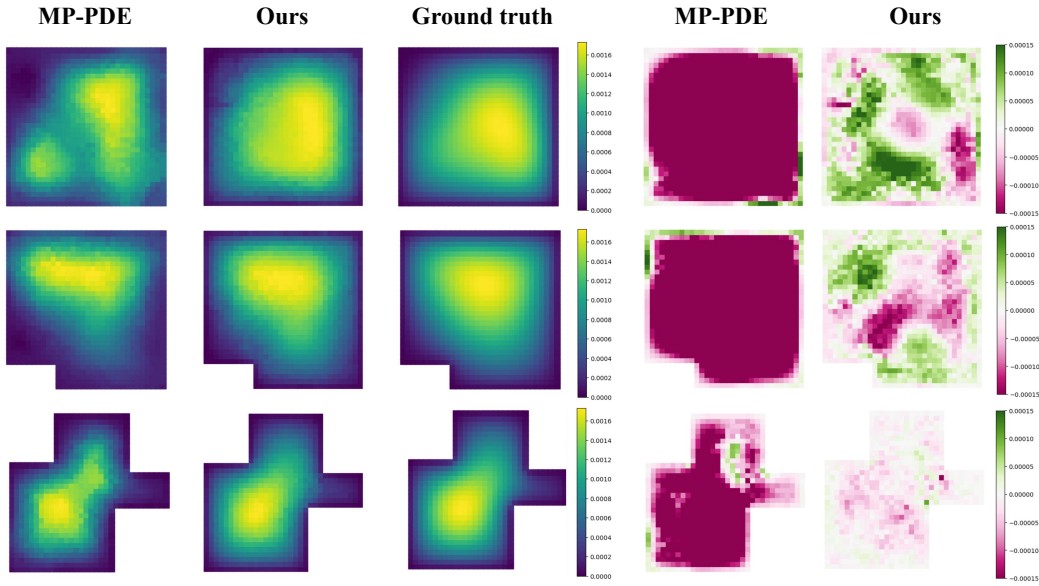

Figure 7: Visualization of prediction and prediction error on data with zero boundary value.

## J   MORE VISUALIZATION ANALYSIS

In this section, we visualize the experimental results on a broader range of experiments. Figure 6 presents the comparison of solution prediction on $64 \times 64$ grid resolution. Figure 7 presents the comparison of solution prediction on data with zero boundary value. Figure 8 presents the qualitative results of training on the 4-Corners dataset and testing on data with various other shapes.

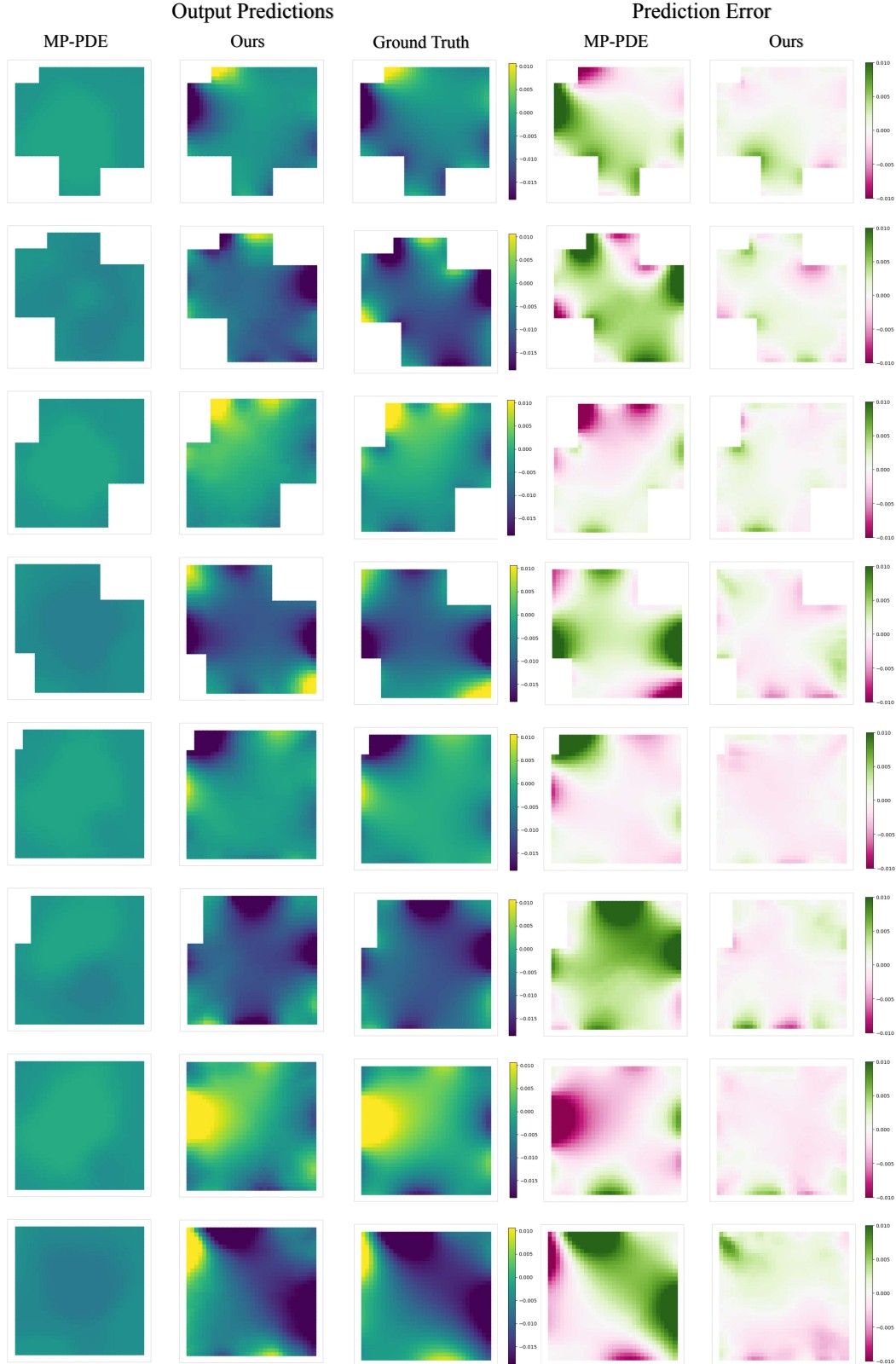

Figure 8: Visualization of prediction and prediction error from 3/2/1/No-Corner dataset, and each has two samples. We render the solution $u$ of the baseline MP-PDE, our BENO and the ground truth in $\Omega$.

Table 8: Performances of our proposed BENO and the compared baselines under Neumann boundary condition, which are trained on 900 4-corners samples and tested on 5 datasets under relative L2 norm and MAE separately. The unit of the MAE metric is $1 \times 10^{-3}$. Bold fonts indicate the best.

| | Train on 4-Corners dataset | | | | | | | | | |
|---|---|---|---|---|---|---|---|---|---|---|
| Test set | 4-Corners | | 3-Corners | | 2-Corners | | 1-Corner | | No-Corner | |
| Metric | L2 | MAE | L2 | MAE | L2 | MAE | L2 | MAE | L2 | MAE |
| GKN | 1.0118± 0.1031 | 3.7105± 1.0699 | 1.0046± 0.1284 | 3.6013± 1.2937 | 1.0301± 0.1417 | 3.3880± 1.0127 | 1.0025± 0.1326 | 3.3675± 0.9112 | 0.9827± 0.1270 | 3.5691± 1.2194 |
| FNO | 1.0547± 0.1643 | 4.0316± 0.8953 | 1.0587± 0.1761 | 4.0219± 0.8210 | 1.0519± 0.1822 | 4.0308± 0.8369 | 1.0533± 0.1782 | 4.0276± 0.8554 | 1.0549± 0.1842 | 4.0417± 0.8063 |
| GNN-PDE | 1.0105± 0.0898 | 2.3685± 0.6933 | 0.9907± 0.1054 | 2.5474± 0.8863 | 1.0132± 0.1208 | 2.7348± 0.8461 | 0.9821± 0.1225 | 2.9824± 0.8106 | 0.9711± 0.1071 | 3.4930± 1.1110 |
| MP-PDE | 1.0070± 0.0813 | 2.3595± 0.6941 | 0.9895± 0.0973 | 2.5480± 0.8955 | 1.0134± 0.1120 | 2.7345± 0.8393 | 0.9782± 0.1240 | 2.9679± 0.7958 | 0.9670± 0.1164 | 3.4807± 1.1143 |
| **BENO (ours)** | **0.3568± 0.0988** | **0.8311± 0.2864** | **0.4201± 0.1170** | **1.0814± 0.3938** | **0.5020± 0.1648** | **1.3918± 0.5454** | **0.5074± 0.1422** | **1.5676± 0.4815** | **0.5221± 0.1474** | **1.8649± 0.5472** |

Table 9: Performances of our proposed BENO and the compared baselines under Neumann boundary condition, which are trained on 900 mixed samples (180 samples each from 5 datasets) and tested on 5 datasets under relative L2 norm and MAE separately. The unit of the MAE metric is $1 \times 10^{-3}$. Bold fonts indicate the best.

| | Train on mixed datasets | | | | | | | | | |
|---|---|---|---|---|---|---|---|---|---|---|
| Test set | 4-Corners | | 3-Corners | | 2-Corners | | 1-Corner | | No-Corner | |
| Metric | L2 | MAE | L2 | MAE | L2 | MAE | L2 | MAE | L2 | MAE |
| GKN | 1.0578± 0.1859 | 3.8992± 1.2048 | 1.0399± 0.2116 | 3.6554± 1.2172 | 1.0975± 0.1912 | 3.5980± 0.9631 | 1.0649± 0.3096 | 3.6030± 0.9310 | 1.0373± 0.2210 | 3.7000± 1.0831 |
| FNO | 1.0426± 0.0917 | 3.5817± 0.8212 | 1.0408± 0.0925 | 3.6187± 0.8338 | 1.0548± 0.1239 | 3.6338± 0.8042 | 1.0592± 0.1065 | 3.6531± 0.8448 | 1.0575± 0.1019 | 3.6498± 0.8239 |
| GNN-PDE | 0.9999± 0.0008 | 2.3648± 0.7703 | 1.0000± 0.0010 | 2.6404± 0.9250 | 0.9999± 0.0010 | 2.7425± 0.9808 | 1.0000± 0.0010 | 3.1458± 0.9672 | 1.0001± 0.0010 | 3.7167± 1.3370 |
| MP-PDE | 1.0245± 0.1048 | 2.3973± 0.7015 | 0.9989± 0.1277 | 2.5510± 0.8717 | 1.0277± 0.1399 | 2.7722± 0.8091 | 0.9940± 0.1543 | 2.9998± 0.7781 | 0.9731± 0.1414 | 3.4930± 1.0867 |
| **BENO (ours)** | **0.4237± 0.1237** | **1.0114± 0.4165** | **0.3970± 0.1277** | **1.0378± 0.4221** | **0.3931± 0.1347** | **1.0881± 0.3993** | **0.3387± 0.1279** | **1.0520± 0.4253** | **0.3344± 0.1171** | **1.2261± 0.4467** |

## K    EXPERIMENTS ON NEUMANN BOUNDARY

In this section, we consider to solve the Poisson equation with Neumann boundary conditions using our proposed BENO. In the context of Neumann boundary conditions, the equation takes the form:

$$
\begin{aligned}
\nabla^2 u(x,y) &= f(x,y), \quad \forall (x,y) \in \Omega, \\
\frac{\partial u(x,y)}{\partial n} &= g(x,y), \quad \forall (x,y) \in \partial\Omega,
\end{aligned}
\tag{18}
$$

where $f$ represents the source term, $n$ typically represents the unit normal vector perpendicular to the boundary surface, and $g$ specifies the prescribed rate of change normal to the boundary $\partial\Omega$. The challenge in solving Poisson's equation with Neumann boundary conditions lies in the proper treatment of the boundary derivative term, which requires sophisticated numerical schemes to approximate accurately.

Specifically, the model is trained exclusively on a dataset consisting of 900 4-corners samples. The robustness and generalizability of our approach were then evaluated on 5 different test datasets, which represent various boundary configurations encountered in practical applications. Each dataset is constructed to challenge the model with different boundary complexities.

The results are shown in Table 8 and Table 9. Our proposed BENO still demonstrates superior performance across all test datasets in comparison to the baselines, including GNN-PDE, and MP-PDE models. Particularly, BENO achieves the lowest MAE and relative L2 norm scores in the majority of the scenarios. In Table 8, when tested on the 4-Corners dataset, BENO exhibits an

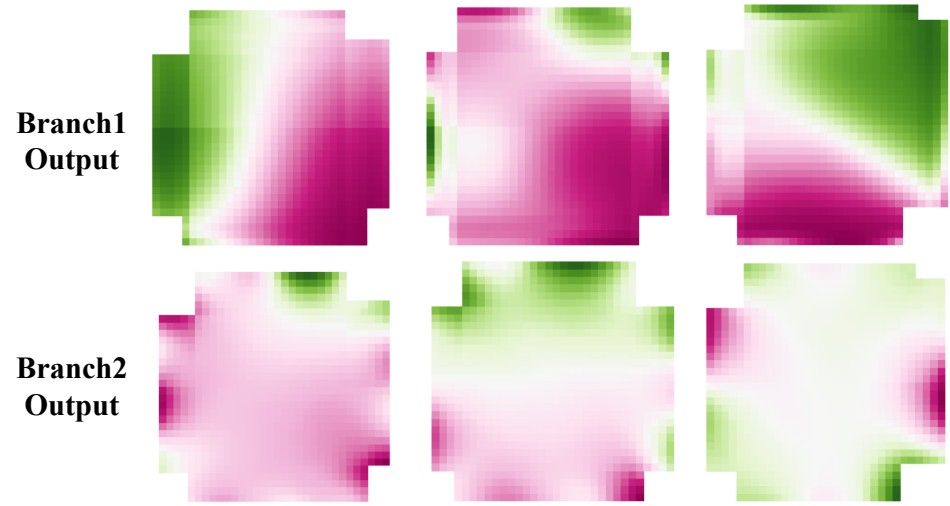

Figure 9: Visualization of the output from 2 GNN branches.

Table 10: Performances of our proposed BENO and the compared baselines on Darcy flow, which are trained on 900 4-corners samples and tested on 5 datasets under relative L2 norm and MAE separately. The unit of the MAE metric is $1 \times 10^{-3}$. Bold fonts indicate the best.

| | Train on 4-Corners dataset | | | | | | | | | |
|---|---|---|---|---|---|---|---|---|---|---|
| Test set | 4-Corners | | 3-Corners | | 2-Corners | | 1-Corner | | No-Corner | |
| Metric | L2 | MAE | L2 | MAE | L2 | MAE | L2 | MAE | L2 | MAE |
| MP-PDE | 0.5802± 0.1840 | 0.3269± 0.2085 | 0.5332± 0.1742 | 0.4652± 0.2999 | 0.6197± 0.1709 | 0.6307± 0.3282 | 0.6906± 0.1432 | 0.8469± 0.4087 | 0.7406± 0.1271 | 1.0906± 0.3949 |
| **BENO (ours)** | **0.2431**± **0.0895** | **0.1664**± **0.0773** | **0.2542**± **0.1252** | **0.2150**± **0.1270** | **0.2672**± **0.1497** | **0.2585**± **0.1313** | **0.2466**± **0.1405** | **0.3091**± **0.2350** | **0.2366**± **0.1104** | **0.3591**± **0.2116** |

L2 norm of 0.3568 and an MAE of 0.8311, outperforming all other methods and showcasing the effectiveness of our approach under strict 4-corners conditions.

When trained on mixed boundary conditions in Table 9, BENO still maintains the highest accuracy, yielding an relative L2 norm of 0.4237 and an MAE of 1.0114 on the 4-Corners test set, confirming its robustness to varied training conditions. Notably, the improvement is significant in the more challenging No-Corner test set, where BENO's L2 is 0.3344, a remarkable enhancement over the baseline methods. The bolded figures in the tables highlight the instances where BENO outperforms all other models, underscoring the impact of our boundary-embedded techniques.

The consistency of BENO's performance under different boundary conditions underscores its potential for applications in computational physics where such scenarios are prevalent. Besides, the experimental outcomes affirm the efficacy of BENO in handling complex boundary problems in the context of PDEs. It is also worth noting that the BENO model not only improves the prediction accuracy but also exhibits a significant reduction in error across different test cases, which is critical for high-stakes applications such as numerical simulation in engineering and physical sciences.

## L  EXPERIMENTS ON DARCY FLOW

In this section, we consider the solution of the Darcy flow using our proposed BENO approach. The 2-d Darcy flow is a second-order linear elliptic equation of the form

$$\begin{aligned} \nabla \cdot (\kappa(x,y)\nabla u(x,y)) &= f(x,y), \quad \forall (x,y) \in \Omega, \\ u(x,y) &= g(x,y), \quad \forall (x,y) \in \partial\Omega, \end{aligned} \tag{19}$$

where the coefficients $\kappa$ is generated by taking a linear combination of smooth basis function in the solution domain. The coefficients of the linear combination of these basis functions is taken from

uniform distribution of random numbers. Dirichlet boundary condition is imposed along the boundary $\partial\Omega$ using the function $g$ which is sufficiently smooth. The objective of BENO is to map from the coefficient $\kappa$ to solution $u$ of the PDE in Equation 19.

The model was exclusively trained on a dataset comprised of 900 samples, each featuring 4-corner configurations. To assess the robustness and adaptability of our method, we conduct evaluations on five distinct test datasets. These datasets are deliberately chosen to represent a variety of boundary conditions commonly encountered in real-world applications, with each one designed to present the model with different levels of boundary complexity. The outcomes of these evaluations are detailed in Table 10. Our proposed BENO model consistently outperforms the best baseline across all test datasets. Notably, BENO achieves the lowest Mean MAE and relative L2 norm in the majority of these scenarios. This performance underscores the effectiveness of our approach, particularly under the stringent conditions of 4-corner boundaries.

## M    VISUALIZATION OF TWO BRANCHES

In this section, the visualized outputs of two distinct branches offer a deeper insight into our model's functionality. Branch1, with the boundary input set to zero, is posited to approximate the impact emanating from the interior, while Branch2, nullifying the interior inputs, is conjectured to capture the boundary's influence on the interior. The observations from Figure 9 lend credence to our hypothesis, indicating a discernible delineation of roles between the two branches.

Extending this analysis, we further postulate that the interplay between Branch1 and Branch2 is critical for accurately modeling the PDE solution landscape. The synergy of these branches, as evidenced in our results, showcases a composite model that effectively balances the intricate boundary-interior dynamics. This balance is crucial in situations where boundary conditions significantly dictate the behavior of the system, further emphasizing the robustness and adaptability of our model. The innovative dual-branch strategy presents a promising avenue for enhancing the interpretability and precision of PDE solutions in complex domains.

## N    HYPER-PARAMETER LIST

Table 11: Hyper-parameters Configuration

| Hyper-parameter Name | Hyper-parameter Value |
|---|---|
| Boundary Dimension | 128 |
| Node Dimension | 128 |
| Edge Dimension | 128 |
| Epochs | 1000 |
| Learning Rate | 5e-05 |
| MLP Layers in Eq. 7 | 3 |
| Nearest Node Number $K$ | 8 |
| Message Passing Steps $T$ | 5 |
| Transformer Layers | 1 |
| Number of Attention Head | 2 |
| Number of iterations for the first restart | 16 |
| Scheduler | CosineAnnealingWarmRestarts |
| Activation Function | Sigmoid Linear Unit (SiLU) |
| GPU Device | Nvidia A100 GPU |

