# OpenReview forum: "BENO: Boundary-embedded Neural Operators for Elliptic PDEs"
_ICLR.cc/2024/Conference — ICLR 2024 poster_

### Official Review · Reviewer_113v · 2023-10-28

**Soundness:** 3 good
**Presentation:** 3 good
**Contribution:** 2 fair
**Rating:** 6
**Confidence:** 4

**Summary:**

The paper presents a method that involves conditioning a Message Passing Neural Network (MPNN) with boundary conditions to predict solutions for the Poisson equation. The approach involves training a model to learn the representation of input boundary conditions using a Transformer encoder. Subsequently, this representation is integrated into two message-passing neural networks, which work together to predict the solution. The model can condition the networks with boundaries of different shapes and values.

**Strengths:**

- The paper tackles the interesting problem of conditioning neural networks with varying boundary conditions and offers a reasonable solution for encoding both the boundary condition and its shape.
- The paper is relatively well-written, providing comprehensive details on the model and the experiments. I appreciate the attention to aligning prior information given to different baselines.

**Weaknesses:**

- Several modeling choices require further clarification through ablation studies:
  - It remains unclear to what extent the separation of the model into two terms is necessary without further empirical proof. Notably, both Branch 1 and Branch 2 share an identical architecture, differing only in their input, which comprises the forcing term and the boundary condition.
  - As illustrated in Figure 1, it seems that the encoding of boundary conditions is achieved using a shared transformer, while the forcing term employs two distinct MPNNs. Is this particular choice of architecture a necessary one?
- Despite the acknowledged disadvantages of concatenating the boundary conditions alongside the other inputs, it is indeed surprising to see the catastrophic failure of recent advances, such as MP-PDE, as illustrated, especially in Figure 5. The authors have mentioned that the learning rate and the weight decay factor are fixed for all experiments through grid search, but it remains unclear with which model these hyperparameters are searched for. Furthermore, aligning optimization hyperparameters across all architectures may benefit certain models but harm the performance of others, and this aspect should be considered. Therefore, more justification should be issued to prove that other baselines do not work at all to support more concretely the claim of a performance gain.
- From the visualization, it appears that there are visible artifacts aligned with boundaries. These artifacts may be introduced by including $(dx_i, dy_i)$ in the node embedding. This raises questions about the necessity of incorporating $(dx_i, dy_i)$, as the Transform already appears to serve in learning a representation of the boundary shape.

----------------------------------------------
Post-rebuttal update: Having reviewed your explanations regarding the clarification of your architectural choices and engaging in discussions with other reviewers, most of my concerns have been addressed. Consequently, I have adjusted my rating to 6.

**Questions:**

- What motivates the use of different implementations for encoding the boundary condition (with Transformer) and the forcing term (with MPNNs)? Even if the goal is to learn a global representation for the boundary condition, couldn't MPNN achieve the same objective?
- How are the 'distances to boundary $(dx_i, dy_i)$' defined? It appears to be more like an offset w.r.t the closest point on the boundary. If there are multiple closest points, how is the choice made?

---

> ### Author Response · Authors · 2023-11-22
> **Response to Reviewer 113v (1)**
>
> > Q1: It remains unclear to what extent the separation of the model into two terms is necessary without further empirical proof. Notably, both Branch 1 and Branch 2 share an identical architecture, differing only in their input, which comprises the forcing term and the boundary condition.
> >
>
> A1: Thanks for raising the question about different branches. Branch 1, with the boundary input set to zero, is posited to approximate the impact emanating from the interior, while Branch 2, nullifying the interior inputs, is conjectured to capture the boundary's influence on the interior. Specifically, we visualize the outputs of two branches which can fully validate our idea. The visualization is shown in the **Appendix** **M** in the updated manuscript, which indicates a discernible delineation of roles between the two branches.
>
> Besides, you can also refer to **BENO w/o. D in Table 6 Sec. 4.4** of the updated manuscript to see the effectiveness of our dual branch architecture
>
> > Q2: As illustrated in Figure 1, it seems that the encoding of boundary conditions is achieved using a shared transformer, while the forcing term employs two distinct MPNNs. Is this particular choice of architecture a necessary one?
> >
>
> A2: Thanks for highlighting the importance of examining the architecture of Transformers. In response, we have conducted an ablation study where we employed the MPNN architecture to encode boundary information, as an alternative to the Transformer model. The results of this study are presented below. They demonstrate that the MPNN architecture is less effective than the Transformer in encoding global boundary information, underscoring the superiority of the latter in this context.
>
> | Test set | 4-Corners L2 | 4-Corners MAE | 3-Corners L2 | 3-Corners MAE | 2-Corners L2 | 2-Corners MAE | 1-Corner L2 | 1-Corner MAE | No-Corner L2 | No-Corner MAE |
> | --- | --- | --- | --- | --- | --- | --- | --- | --- | --- | --- |
> | BENNO w/o dx,dy | 0.5985±0.1111 | 1.8584±0.6242 | 0.6175±0.1105 | 2.0114±0.6173 | 0.6255±0.1122 | 2.0424±0.6686 | 0.6145±0.1172 | 1.8982±0.6453 | 0.6445±0.1103 | 2.3310±0.7341 |
> | BENO (ours) | 0.3523±0.1245 | 0.9650±0.3131 | 0.4308±0.1994 | 1.2206±0.4978 | 0.4910±0.1888 | 1.4388±0.5227 | 0.5416±0.2133 | 1.4529±0.4626 | 0.5542±0.1952 | 1.7481±0.5394 |
>
> > Q3: Despite the acknowledged disadvantages of concatenating the boundary conditions alongside the other inputs, it is indeed surprising to see the catastrophic failure of recent advances, such as MP-PDE, as illustrated, especially in Figure 5. The authors have mentioned that the learning rate and the weight decay factor are fixed for all experiments through grid search, but it remains unclear with which model these hyperparameters are searched for. Furthermore, aligning optimization hyperparameters across all architectures may benefit certain models but harm the performance of others, and this aspect should be considered. Therefore, more justification should be issued to prove that other baselines do not work at all to support more concretely the claim of a performance gain.
> >
>
> A3: Thanks for the comment. In pursuit of a more equitable comparison, we have adopted additional strategies to enhance the baseline's performance. Firstly, we augmented the original MP-PDE with a virtual node linked to all boundaries (which has demonstrated to improve the performance, as is also supported by our experiments). Subsequently, we conducted an optimal hyper-parameter search to fine-tune the hyper-parameters. The ensuing results are presented below. Despite these efforts, a notable performance disparity remains when contrasted with our BENO approach. The baseline, even with these enhancements, does not successfully resolve the issue at hand.
>
> | Learning rate | Relative L2 | MAE |
> | --- | --- | --- |
> | 1e-5 | 0.9324±0.1521 | 2.9755±1.0400 |
> | 5e-5 | 0.8684±0.1471 | 2.7438±1.0008 |
> | 1e-4 | 0.8560±0.1610 | 2.6336±1.0063 |
> | 5e-4 | 1.0036±0.0121 | 3.1454±0.8793 |
>
> | MLP layers | Relative L2 | MAE |
> | --- | --- | --- |
> | 1 | 0.9007±0.1381 | 2.8799±0.9726 |
> | 2 | 0.8915±0.1561 | 2.8696±1.0253 |
> | 3 | 0.8684±0.1471 | 2.7438±1.0008 |
> | 4 | 0.8879±0.1312 | 2.8139±0.9726 |

---

> ### Author Response · Authors · 2023-11-22
> **Response to Reviewer 113v (2)**
>
> > Q4: From the visualization, it appears that there are visible artifacts aligned with boundaries. These artifacts may be introduced by including in the node embedding. This raises questions about the necessity of incorporating (dx, dy), as the Transform already appears to serve in learning a representation of the boundary shape
> >
>
> A4: Thanks for the comment. We have added ablation study that removes the dx, dy to validate its effectiveness, which is shown as follows. It is obvious that this feature is helpful for our PDE solving.
>
> | Test set | 4-Corners L2 | 4-Corners MAE | 3-Corners L2 | 3-Corners MAE | 2-Corners L2 | 2-Corners MAE | 1-Corner L2 | 1-Corner MAE | No-Corner L2 | No-Corner MAE |
> | --- | --- | --- | --- | --- | --- | --- | --- | --- | --- | --- |
> | BENNO w/o dx,dy | 0.3654±0.1591 | 0.9989±0.3752 | 0.4535±0.2261 | 1.2912±0.5961 | 0.5549±0.2044 | 1.5689±0.5708 | 0.5797±0.2219 | 1.535±0.5039 | 0.5818±0.2219 | 1.7535±0.5547 |
> | BENO (ours) | 0.3523±0.1245 | 0.9650±0.3131 | 0.4308±0.1994 | 1.2206±0.4978 | 0.4910±0.1888 | 1.4388±0.5227 | 0.5416±0.2133 | 1.4529±0.4626 | 0.5542±0.1952 | 1.7481±0.5394 |
>
> > Q5: What motivates the use of different implementations for encoding the boundary condition (with Transformer) and the forcing term (with MPNNs)? Even if the goal is to learn a global representation for the boundary condition, couldn't MPNN achieve the same objective?
> >
>
> A5: Thanks for highlighting the importance of examining the architecture of Transformers. In response, we have conducted an ablation study where we employed the MPNN architecture to encode boundary information, as an alternative to the Transformer model. The results of this study are presented below. They demonstrate that the MPNN architecture is less effective than the Transformer in encoding global boundary information, underscoring the superiority of the latter in this context.
>
> | Test set | 4-Corners L2 | 4-Corners MAE | 3-Corners L2 | 3-Corners MAE | 2-Corners L2 | 2-Corners MAE | 1-Corner L2 | 1-Corner MAE | No-Corner L2 | No-Corner MAE |
> | --- | --- | --- | --- | --- | --- | --- | --- | --- | --- | --- |
> | BENNO w/o dx,dy | 0.5985±0.1111 | 1.8584±0.6242 | 0.6175±0.1105 | 2.0114±0.6173 | 0.6255±0.1122 | 2.0424±0.6686 | 0.6145±0.1172 | 1.8982±0.6453 | 0.6445±0.1103 | 2.3310±0.7341 |
> | BENO (ours) | 0.3523±0.1245 | 0.9650±0.3131 | 0.4308±0.1994 | 1.2206±0.4978 | 0.4910±0.1888 | 1.4388±0.5227 | 0.5416±0.2133 | 1.4529±0.4626 | 0.5542±0.1952 | 1.7481±0.5394 |
>
> > Q6: How are the 'distances to boundary ' defined? It appears to be more like an offset w.r.t the closest point on the boundary. If there are multiple closest points, how is the choice made?
> >
>
> A6: The calculation is based on the nearest point along the boundary. Given that the mesh points are arranged in a regular pattern, there is only one closest point for each calculation.

---

> ### Author Response · Authors · 2023-11-23
> **A gentle reminder: please let us know if you have additional questions**
>
> Dear Reviewer 113v,
>
> We appreciate your time to review and the constructive comments to encourage our work. We want to leave a gentle reminder due to nearing the end of the discussion period.
>
> We have tried to address all your concerns by providing more explanations and results. Please go over our response, and if you have additional questions, please let us know.
>
> Thank you,
>
> the Authors.

---

### Official Review · Reviewer_tbt5 · 2023-10-29

**Soundness:** 3 good
**Presentation:** 3 good
**Contribution:** 3 good
**Rating:** 8
**Confidence:** 4

**Summary:**

The paper proposed a novel neural operator for elliptic equations, particularly Poisson equations with Dirichlet boundary conditions in 2D domain, that has better flexibility to the geometry transformation of the problem domain.

The neural operator named BENO has several advantages over previous methods. No previous method can have train/test space grid independence, evaluation at on observed spatial locations, free-form spatial domain for boundary shape, inhomogeneous boundary condition values, at the same time.

**Strengths:**

**Originality:** The paper has deliberately designed a neural operator for PDE with Dirichlet boundary conditions on free-form geometry domains.

**Quality:** The paper has extensively benchmarked BENO with many existing NOs, and BENO shows great advantages.

**Clarity:** The paper is well organized and clear presented.

**Significance:** To have NO be able to deal with Dirichlet boundary conditions on domains of complex geometry is very important to the community, which can greatly improve the flexibility of NO and the range of scenario where NO can be useful.

**Weaknesses:**

The paper only deal with Poisson equations with Dirichlet boundary conditions, which limits the effect on convincing readers that BENO is superior than baselines.

How about more general elliptic equations?

**Questions:**

None.

---

> ### Author Response · Authors · 2023-11-22
> **Response to Reviewer tbt5**
>
> > Q1: The paper only deal with Poisson equations with Dirichlet boundary conditions, which limits the effect on convincing readers that BENO is superior than baselines
> >
>
> A1:  Thanks for your comment. We have generated new datasets on different boundary condition (i.e., **Neumann BCs**) under our existing setting with different corner cases. Typically boundary conditions can be categorized into Dirichlet and Neumann BCs, or the combination of the two. We believe that the addition of Neumann BCs to the manuscript makes the paper to be applicable to a much wider range of systems. We represent the supplementary experiment results as follows:
>
> - Performances of our BENO and the compared baseline under **Neumann boundary condition** are shown below, which are **trained on 900 4-corners** samples and tested on 5 datasets under relative L2 norm and MAE separately. The whole comparison table with all the baselines is shown in **Appendix L**.
>
> | Test set | 4-Corners L2 | 4-Corners MAE | 3-Corners L2 | 3-Corners MAE | 2-Corners L2 | 2-Corners MAE | 1-Corner L2 | 1-Corner MAE | No-Corner L2 | No-Corner MAE |
> | --- | --- | --- | --- | --- | --- | --- | --- | --- | --- | --- |
> | MP-PDE | 1.0070±0.0813 | 2.3595±0.6941 | 0.9895±0.0973 | 2.5480±0.8955 | 1.0134±0.1120 | 2.7345±0.8393 | 0.9782±0.1240 | 2.9679±0.7958 | 0.9670±0.1164 | 3.4807±1.1143 |
> | BENO (ours) | 0.3568±0.0988 | 0.8311±0.2864 | 0.4201±0.1170 | 1.0814±0.3938 | 0.5020±0.1648 | 1.3918±0.5454 | 0.5074±0.1422 | 1.5676±0.4815 | 0.5221±0.1474 | 1.8649±0.5472 |
> - Performances of our BENO and the compared baseline under **Neumann boundary condition** are shown below, which are **trained on 900 mixed** samples (180 samples each from 5 datasets) and tested on 5 datasets under relative L2 norm and MAE separately. The whole comparison table with all the baselines is shown in **Appendix L**.
>
> | Test set | 4-Corners L2 | 4-Corners MAE | 3-Corners L2 | 3-Corners MAE | 2-Corners L2 | 2-Corners MAE | 1-Corner L2 | 1-Corner MAE | No-Corner L2 | No-Corner MAE |
> | --- | --- | --- | --- | --- | --- | --- | --- | --- | --- | --- |
> | MP-PDE | 1.0245±0.1048 | 2.3973±0.7015 | 0.9989±0.1277 | 2.5510±0.8717 | 1.0277±0.1399 | 2.7722±0.8091 | 0.9940±0.1543 | 2.9998±0.7781 | 0.9731±0.1414 | 3.4930±1.0867 |
> | BENO (ours) | 0.4237±0.1237 | 1.0114±0.4165 | 0.3970±0.1277 | 1.0378±0.4221 | 0.3931±0.1347 | 1.0881±0.3993 | 0.3387±0.1279 | 1.0520±0.4253 | 0.3344±0.1171 | 1.2261±0.4467 |
>
> Q2: How about more general elliptic equations?
>
> A2: Thanks for the comment. We have generated new datasets on **Darcy flow** (mapping from coefficient to solution) under our existing settings. Performances of our BENO and the compared baseline on **Darcy flow** are shown below**,** which are **trained on 900 4-corner** samples (180 samples each from 5 datasets) and tested on 5 datasets under relative L2 norm and MAE separately. The whole comparison table with all the baselines is shown in **Appendix M**.
>
> | Test set | 4-Corners L2 | 4-Corners MAE | 3-Corners L2 | 3-Corners MAE | 2-Corners L2 | 2-Corners MAE | 1-Corner L2 | 1-Corner MAE | No-Corner L2 | No-Corner MAE |
> | --- | --- | --- | --- | --- | --- | --- | --- | --- | --- | --- |
> | MP-PDE | 0.4802±0.1840 | 0.3269±0.2085 | 0.5332±0.1742 | 0.4652±0.2999 | 0.6197±0.1709 | 0.6307±0.3282 | 0.6906±0.1432 | 0.8469±0.4087 | 0.7406±0.1271 | 1.0906±0.3949 |
> | BENO (ours) | 0.2431±0.0895 | 0.1664±0.0773 | 0.2542±0.1252 | 0.2150±0.1270 | 0.2672±0.1497 | 0.2585±0.1313 | 0.2466±0.1405 | 0.3091±0.2350 | 0.2366±0.1104 | 0.3591±0.2116 |

---

> > ### Comment · Reviewer_tbt5 · 2023-11-22
> >
> > Thanks for your reply. I think the additional experiments are very helpful. After reading other reviews, I still think this a good paper that improves neural operators on complex geometry domains. Therefore, my score remains.

---

> > > ### Author Response · Authors · 2023-11-22
> > > **Response to Reviewer tbt5**
> > >
> > > Thanks for your valuable feedback on our paper. We genuinely value your contributions and will ensure that any further suggestions will be carefully incorporated.

---

### Official Review · Reviewer_MqG8 · 2023-11-01

**Soundness:** 3 good
**Presentation:** 2 fair
**Contribution:** 1 poor
**Rating:** 5
**Confidence:** 3

**Summary:**

The existing models for operator learning have struggled to handle complex domains and boundary conditions effectively. To address this challenge, this paper introduces the Boundary-Embedded Neural Operators (BENO) model. BENO focuses on elliptic partial differential equations (PDEs) and employs separate graph neural networks for learning the source term inside the domain and the boundary values at the boundaries. It leverages message passing and transformers to learn the solution operator. Experimental results demonstrate that BENO offers more accurate predictions for areas with diverse boundaries compared to existing models.

### Post-rebuttal
I apologize for the delay in my response. Thank you for addressing most of my inquiries and concerns despite the limited time. I appreciate the swift inclusion of various experiments, which has alleviated my concerns. I hope that my questions and suggestions contribute to making the paper even better.

However, I still have reservations about the generalization of this method. As observed in the current experiments, the domain handled is limited to rectangles with cut corners. I remain skeptical about its applicability to smoother circles or more complex domains, as I originally inquired. Instead of conducting experiments on similar domains, it would be beneficial to showcase its performance on entirely different shapes and more intricate domains. Although the authors mentioned this as future work in their response, it seems that further exploration in this direction is warranted.

Therefore, based on my personal opinion, I will revise my score to a maximum of 5 points. I greatly appreciate your kind and diligent responses to my review.

**Strengths:**

The issue raised in the paper, where existing neural operator models struggle to handle complex domains and challenging boundary conditions, is indeed a significant problem that needs to be addressed. Many existing neural operator models often disregard PDEs and neglect boundary conditions when working with data. Furthermore, the problem posed in the paper is crucial in this field since operator learning is typically performed in straightforward domains. Moreover, the idea of using triangulation to locally partition complex domains is a promising approach to tackling these challenges.

**Weaknesses:**

However, the paper's focus on elliptic PDEs seems somewhat restrictive, especially considering that it primarily deals with the simplest case of the Poisson equation with Dirichlet boundary conditions. This might give the impression that the novelty of the paper is somewhat limited.

For example, can the proposed BENO method be applied to Poisson equations with Neumann boundary conditions or mixed Robin boundary conditions? The current model design appears to be tailored to the Dirichlet boundary conditions, which might be considered quite restrictive. To truly highlight the advantages of this approach, it would be beneficial to demonstrate its applicability to a wider range of PDEs, including parametric PDEs such as convection-diffusion equations or nonlinear Burgers' equations. Moreover, exploring its applicability to time-dependent parabolic PDEs could lead to a more generalized operator learning model.

In this context, the paper experimentally investigates a single type of operator, the one between (f, g) and the solution u. However, it's intriguing to consider whether other types of solution operators could also be learned (for instance, as seen in the FNO paper with Darcy flow, where the coefficient could be an input to the operator, or the initial condition of PDE can be an input).

Furthermore, the experiments in complex domains appear to be limited to similar, complex domains, such as the "5 different corner elliptic dataset." It would be interesting to explore whether the BENO method can be extended and applied to different scenarios, such as domains with circular holes in the center of a square or corners at non-right angles (or more simple, circle domain as in (Lotzsch et al., 2022)). By conducting experiments in more diverse domains and presenting results in a variety of scenarios, the paper could better showcase the novelty and versatility of the BENO method.

Therefore, it would be beneficial to generalize the limitations of BENO, as described in this paper. For instance, in terms of boundary conditions, there are relevant discussions in [1], and graph-based methods have addressed complex domains and a variety of equations and boundary conditions, as seen in [2] and [3], which leveraged Finite Element Method (FEM). Referencing these works might provide valuable insights.

Furthermore, there is some curiosity regarding whether the results of the baseline methods were adequately compared. It appears from Appendix F that the internal and boundary grids for the baseline methods are distinguished using one-hot encoding. It's worth considering if this method provides the fairest basis for comparison. As mentioned in the paper, FNO suggests the use of Fourier continuation for different boundary conditions and domains ('Non-uniform and Non-periodic Geometry' part in [4]). Therefore, it raises the question of whether the paper's approach is indeed the best for addressing complex scenarios. Additionally, there is a model for operator learning called DeepONet [5], which can be applied directly to challenging boundary conditions and complex domains. It would be interesting to know whether the paper conducted a comparison with this model.

*[1] Horie, M., & Mitsume, N. (2022). Physics-Embedded Neural Networks: Graph Neural PDE Solvers with Mixed Boundary Conditions. Advances in Neural Information Processing Systems, 35, 23218-23229.*

*[2] Lienen, M., & Günnemann, S. (2022). Learning the dynamics of physical systems from sparse observations with finite element networks. arXiv preprint arXiv:2203.08852.*

*[3] Lee, J. Y., Ko, S., & Hong, Y. (2023). Finite Element Operator Network for Solving Parametric PDEs. arXiv preprint arXiv:2308.04690.*

*[4] Kovachki, N. B., Li, Z., Liu, B., Azizzadenesheli, K., Bhattacharya, K., Stuart, A. M., & Anandkumar, A. (2023). Neural Operator: Learning Maps Between Function Spaces With Applications to PDEs. J. Mach. Learn. Res., 24(89), 1-97.*

*[5] Lu, L., Meng, X., Cai, S., Mao, Z., Goswami, S., Zhang, Z., & Karniadakis, G. E. (2022). A comprehensive and fair comparison of two neural operators (with practical extensions) based on fair data. Computer Methods in Applied Mechanics and Engineering, 393, 114778.*

**Questions:**

* In the paper, it is mentioned that other methods overfit in challenging boundary conditions, but a more explicit explanation of what this means would be helpful. Are there graphs or figures that can demonstrate this concept?

* It would be beneficial to provide a clearer explanation of what the red and orange parts in Figure 1 represent. What does the red part signify, and what about the orange part? How do they differ?

* Regarding Table 1, as mentioned in the Strengths section, is it impossible for FNO to handle 5.? Is it also impossible for GKN to handle 5.? Is there no prior research in the field of operator learning models that can handle 5.?

* In the Problem Setup (2. Problem Setup), it is not clear whether BENO can be applied to 3D or higher-dimensional domains. While the use of graph neural networks suggests that it might be possible, further explanation would be helpful.

* In the context of 3.1 Motivation, where branch1 is interpreted as the interior and branch2 as the exterior from a Green function perspective, it's important to know whether the output of Branch1 precisely reaches 0 at the boundary and whether the output of Branch2 reaches exactly "g" at the boundary. Is this a process of approximating "g," or is the model capable of obtaining the exact boundary values of "g"?

* In Section 3.2, a more detailed explanation of the definition of $\mathcal{E}_{kn}$ would be appreciated. Furthermore, the part around this section involving attributes is somewhat challenging to comprehend in written form. It might be a good idea to exclude this part if it ultimately relates to the explanation in Section 3.3.2.

* Are dx, dy, and dc in Section 3.3.1 uniquely determined for each node? Also, why is dc necessary?

* On page 5, is "B" ultimately equal to "B^1"? It would be beneficial to provide a more detailed explanation of where "t" changes and what its range is.

* The definition of MAE for Table 2 is present in the appendix but not in the main text.

* In Section 4.3.2, the explanation of training at 32x32 and testing at 64x64, is this capability attributed to BENO's use of graph neural networks, or are there other features enabling this super-resolution? A more detailed explanation would be appreciated.

* In Appendix K, when looking at Figure 6, it appears that the results for "Ours" are different from other solution profiles, forming square blocks with discontinuities. What is the reason or cause behind this phenomenon?

---

> ### Author Response · Authors · 2023-11-22
> **Response to Reviewer MqG8 (1)**
>
> > Q1: However, the paper's focus on elliptic PDEs seems somewhat restrictive, especially considering that it primarily deals with the simplest case of the Poisson equation with Dirichlet boundary conditions. This might give the impression that the novelty of the paper is somewhat limited.
> >
>
> A1:  Despite the focus on elliptic PDEs, we believe this is an important and critical research that can be applicable to a wide range of natural phenomena and engineering applications.
>
> - **Importance of boundary conditions for any PDE systems.** It is known that any PDEs can be characterized as parabolic, hyperbolic, and elliptic PDEs, which further can be categorized as initial-boundary value problem (IBVP), and initial value problem (IVP) and boundary value problem (BVP), respectively. For any of these PDEs, the treatment of boundary conditions plays a critical role in accurately modeling natural phenomena and engineering applications. While we focus on the application of BENO on elliptic PDEs, we believe that the boundary condition treatment and incorporation into the overall solver is applicable to other PDE systems.
> - **Applicability of elliptic PDEs to the real-world phenomena.** The utility of elliptic PDEs is not limited to steady problems, e.g., we can obtain unsteady solution of nonlinear PDEs by repeated solution of elliptic PDEs. For instance, the pressure-Poisson equation (which is an elliptic PDE) for incompressible flows can be applied to unsteady phenomena. Various plasma simulations uses Poisson solver that is time dependent. Helmholtz equation is used in a variety of phenomena, including electromagnetic radiation, seismology, and acoustics. These systems can have complex geometries, e.g., airfoil, propulsion devices, plasma sources for semiconductor manufacturing, with different types of materials (e.g., dielectric, conducting, catalytic, chemically reactive) and inhomogeneous boundary values (e.g., dependent on space and time). Thus, we believe that demonstration of BENO on elliptic PDEs has immense impact to a large number of applications. This introduction is updated to emphasize this point. Besides, we have revised the manuscript to delete the statements that elliptic PDEs are only applicable for steady-state phenomena.
>
> Furthermore, to validate the generalization ability of our BENO, we **also supplement more experiments** on other scenarios in the following part.
>
> Besides, to validate the generalization ability of our BENO, we **also supplement more experiments** on other scenarios in the following part.

---

> ### Author Response · Authors · 2023-11-22
> **Response to Reviewer MqG8 (2)**
>
> > Q2: For example, can the proposed BENO method be applied to Poisson equations with Neumann boundary conditions or mixed Robin boundary conditions? The current model design appears to be tailored to the Dirichlet boundary conditions, which might be considered quite restrictive.
> >
>
> A2:  Thanks for your comment. We have generated new datasets on different boundary condition (i.e., **Neumann BCs**) under our existing setting with different corner cases. Typically boundary conditions can be categorized into Dirichlet and Neumann BCs, or the combination of the two. We believe that the addition of Neumann BCs to the manuscript makes the paper to be applicable to a much wider range of systems. We represent the supplementary experiment results as follows:
>
> - Performances of our BENO and the compared baseline under **Neumann boundary condition** are shown below, which are **trained on 900 4-corners** samples and tested on 5 datasets under relative L2 norm and MAE separately. The whole comparison table with all the baselines is shown in **Appendix L**.
>
> | Test set | 4-Corners L2 | 4-Corners MAE | 3-Corners L2 | 3-Corners MAE | 2-Corners L2 | 2-Corners MAE | 1-Corner L2 | 1-Corner MAE | No-Corner L2 | No-Corner MAE |
> | --- | --- | --- | --- | --- | --- | --- | --- | --- | --- | --- |
> | MP-PDE | 1.0070±0.0813 | 2.3595±0.6941 | 0.9895±0.0973 | 2.5480±0.8955 | 1.0134±0.1120 | 2.7345±0.8393 | 0.9782±0.1240 | 2.9679±0.7958 | 0.9670±0.1164 | 3.4807±1.1143 |
> | BENO (ours) | 0.3568±0.0988 | 0.8311±0.2864 | 0.4201±0.1170 | 1.0814±0.3938 | 0.5020±0.1648 | 1.3918±0.5454 | 0.5074±0.1422 | 1.5676±0.4815 | 0.5221±0.1474 | 1.8649±0.5472 |
> - Performances of our BENO and the compared baseline under **Neumann boundary condition** are shown below, which are **trained on 900 mixed** samples (180 samples each from 5 datasets) and tested on 5 datasets under relative L2 norm and MAE separately. The whole comparison table with all the baselines is shown in **Appendix L**.
>
> | Test set | 4-Corners L2 | 4-Corners MAE | 3-Corners L2 | 3-Corners MAE | 2-Corners L2 | 2-Corners MAE | 1-Corner L2 | 1-Corner MAE | No-Corner L2 | No-Corner MAE |
> | --- | --- | --- | --- | --- | --- | --- | --- | --- | --- | --- |
> | MP-PDE | 1.0245±0.1048 | 2.3973±0.7015 | 0.9989±0.1277 | 2.5510±0.8717 | 1.0277±0.1399 | 2.7722±0.8091 | 0.9940±0.1543 | 2.9998±0.7781 | 0.9731±0.1414 | 3.4930±1.0867 |
> | BENO (ours) | 0.4237±0.1237 | 1.0114±0.4165 | 0.3970±0.1277 | 1.0378±0.4221 | 0.3931±0.1347 | 1.0881±0.3993 | 0.3387±0.1279 | 1.0520±0.4253 | 0.3344±0.1171 | 1.2261±0.4467 |
>
> > Q3: In this context, the paper experimentally investigates a single type of operator, the one between (f, g) and the solution u. However, it's intriguing to consider whether other types of solution operators could also be learned (for instance, as seen in the FNO paper with Darcy flow, where the coefficient could be an input to the operator, or the initial condition of PDE can be an input).
> >
>
> A3: Thanks for the comment. We have generated new datasets on **Darcy flow** (mapping from coefficient to solution) under our existing settings with different corner casess. Performances of our BENO and the compared baseline on **Darcy flow** are shown below**,** which are **trained on 900 4-corner** samples (180 samples each from 5 datasets) and tested on 5 datasets under relative L2 norm and MAE separately. The whole comparison table with all the baselines is shown in **Appendix M**.
>
> | Test set | 4-Corners L2 | 4-Corners MAE | 3-Corners L2 | 3-Corners MAE | 2-Corners L2 | 2-Corners MAE | 1-Corner L2 | 1-Corner MAE | No-Corner L2 | No-Corner MAE |
> | --- | --- | --- | --- | --- | --- | --- | --- | --- | --- | --- |
> | MP-PDE | 0.4802±0.1840 | 0.3269±0.2085 | 0.5332±0.1742 | 0.4652±0.2999 | 0.6197±0.1709 | 0.6307±0.3282 | 0.6906±0.1432 | 0.8469±0.4087 | 0.7406±0.1271 | 1.0906±0.3949 |
> | BENO (ours) | 0.2431±0.0895 | 0.1664±0.0773 | 0.2542±0.1252 | 0.2150±0.1270 | 0.2672±0.1497 | 0.2585±0.1313 | 0.2466±0.1405 | 0.3091±0.2350 | 0.2366±0.1104 | 0.3591±0.2116 |

---

> ### Author Response · Authors · 2023-11-22
> **Response to Reviewer MqG8 (3)**
>
> > Q4: Furthermore, the experiments in complex domains appear to be limited to similar, complex domains, such as the "5 different corner elliptic dataset." It would be interesting to explore whether the BENO method can be extended and applied to different scenarios, such as domains with circular holes in the center of a square or corners at non-right angles (or more simple, circle domain as in (Lotzsch et al., 2022)). By conducting experiments in more diverse domains and presenting results in a variety of scenarios, the paper could better showcase the novelty and versatility of the BENO method.
> >
>
> A4: Thanks for the positive feedback. Previous work i.e., GNN-PDE, utilizes methodology on circular holes but may have neglected the treatment of complex physical boundary conditions (including combination of Dirichlet and Neumann boundaries, which can be inhomogeneous), which we believe is the key innovativeness of our work. Generalization of the method (e.g., circular, spherical, unstructured meshes) is reserved for future work.
>
> > Q5: Therefore, it would be beneficial to generalize the limitations of BENO, as described in this paper. For instance, in terms of boundary conditions, there are relevant discussions in [1], and graph-based methods have addressed complex domains and a variety of equations and boundary conditions, as seen in [2] and [3], which leveraged Finite Element Method (FEM). Referencing these works might provide valuable insights.
> >
>
> A5: Thanks for pointing out these references. We have added these references in the revised version of 5. Related Work section for a boarder audience.
>
> > Q6: Furthermore, there is some curiosity regarding whether the results of the baseline methods were adequately compared. It appears from Appendix F that the internal and boundary grids for the baseline methods are distinguished using one-hot encoding. It's worth considering if this method provides the fairest basis for comparison. As mentioned in the paper, FNO suggests the use of Fourier continuation for different boundary conditions and domains ('Non-uniform and Non-periodic Geometry' part in [4]). Therefore, it raises the question of whether the paper's approach is indeed the best for addressing complex scenarios. Additionally, there is a model for operator learning called DeepONet [5], which can be applied directly to challenging boundary conditions and complex domains. It would be interesting to know whether the paper conducted a comparison with this model.
> >
>
> A6: Thank you for your comment. In pursuit of a more equitable comparison, we have adopted additional strategies to enhance the baseline's performance. Firstly, we augmented the original MP-PDE with a virtual node linked to all boundaries. Subsequently, we conducted an optimal hyper-parameter search to fine-tune the hyper-parameters. The ensuing results are presented below. Despite these efforts, a notable performance disparity remains when contrasted with our BENO approach. The baseline, even with these enhancements, does not successfully resolve the issue at hand.
>
> | Learning rate | Relative L2 | MAE |
> | --- | --- | --- |
> | 1e-5 | 0.9324±0.1521 | 2.9755±1.0400 |
> | 5e-5 | 0.8684±0.1471 | 2.7438±1.0008 |
> | 1e-4 | 0.8560±0.1610 | 2.6336±1.0063 |
> | 5e-4 | 1.0036±0.0121 | 3.1454±0.8793 |
>
> | MLP layers | Relative L2 | MAE |
> | --- | --- | --- |
> | 1 | 0.9007±0.1381 | 2.8799±0.9726 |
> | 2 | 0.8915±0.1561 | 2.8696±1.0253 |
> | 3 | 0.8684±0.1471 | 2.7438±1.0008 |
> | 4 | 0.8879±0.1312 | 2.8139±0.9726 |
>
> And it should be noted that the boundary conditions utilized in the related work are all homogeneous. We have also added the ****DeepONet**** as the baseline, and implement experiments under our main setting, which is trained on 900 4-corners samples and tested on 5 datasets. The results are shown as below and it is obvious that our BENO surpass the baseline by a large margin.
>
> | Test set | 4-Corners L2 | 4-Corners MAE | 3-Corners L2 | 3-Corners MAE | 2-Corners L2 | 2-Corners MAE | 1-Corner L2 | 1-Corner MAE | No-Corner L2 | No-Corner MAE |
> | --- | --- | --- | --- | --- | --- | --- | --- | --- | --- | --- |
> | DeepONet | 1.0002±0.1029 | 2.4815±0.3216 | 1.0002±0.1277 | 2.4768±0.3519 | 1.0005±0.1487 | 2.5216±0.3078 | 1.0117±0.1762 | 2.7018±0.4219 | 1.0075±0.1923 | 2.5529±0.3116 |
> | BENO (ours) | 0.3523±0.1245 | 0.9650±0.3131 | 0.4308±0.1994 | 1.2206±0.4978 | 0.4910±0.1888 | 1.4388±0.5227 | 0.5416±0.2133 | 1.4529±0.4626 | 0.5542±0.1952 | 1.7481±0.5394 |

---

> ### Author Response · Authors · 2023-11-22
> **Response to Reviewer MqG8 (4)**
>
> > Q7: In the paper, it is mentioned that other methods overfit in challenging boundary conditions, but a more explicit explanation of what this means would be helpful. Are there graphs or figures that can demonstrate this concept?
> >
>
> A7: Thank you for bringing attention to the issue regarding the failure of other methods. We have updated our manuscript to reflect a more precise terminology, changing 'overfitting' to 'extremely hard to learn'. This revision enhances the accuracy of our statements. The evidence for this adjustment is evident from **Figure 4 in Section 4.2** and **Figure 5 in Appendix I.3**. These figures clearly illustrate that the baseline methods struggle to converge the loss during both training and testing phases. Additionally, there is a markedly high prediction error when compared to the ground truth, further underscoring the challenges these methods face.
> > Q8: It would be beneficial to provide a clearer explanation of what the red and orange parts in Figure 1 represent. What does the red part signify, and what about the orange part? How do they differ?
> >
>
> A8: The red and orange hues in Figure 1 are chosen from the color-bar for our graphical representation, which effectively illustrates the boundary values. The redder the area, the higher the boundary value it represents, whereas the more orange the area, the lower the boundary value. We have added this information in the caption of Figure 1 in the updated manuscript.
>
> > Q9: Regarding Table 1, as mentioned in the Strengths section, is it impossible for FNO to handle 5.? Is it also impossible for GKN to handle 5.? Is there no prior research in the field of operator learning models that can handle 5.?
> >
>
> A9: To the best of our knowledge, no work has stepped in solving elliptic PDEs with complex boundary shape and inhomogeneous boundary value together. As for all of the baselines, we utilize the same information as input for fair comparison and it is obvious that existing vanilla baselines cannot deal with the complex setting, which just demonstrates that a new architecture design is needed.
>
> > Q10: In the Problem Setup (2. Problem Setup), it is not clear whether BENO can be applied to 3D or higher-dimensional domains. While the use of graph neural networks suggests that it might be possible, further explanation would be helpful.
> >
>
> A10: We have refined our setup in **Section 2** to enable its generalization to more diverse scenarios.
>
> > Q11: In the context of 3.1 Motivation, where branch1 is interpreted as the interior and branch2 as the exterior from a Green function perspective, it's important to know whether the output of Branch1 precisely reaches 0 at the boundary and whether the output of Branch2 reaches exactly "g" at the boundary. Is this a process of approximating "g," or is the model capable of obtaining the exact boundary values of "g"?
> >
>
> A11: Thanks for raising the question about different branches. Branch 1, with the boundary input set to zero, is posited to approximate the impact emanating from the interior, while Branch 2, nullifying the interior inputs, is conjectured to capture the boundary's influence on the interior. Specifically, we visualize the outputs of two branches which can fully validate our idea. The visualization is shown in the **Appendix** **N**, which indicates a discernible delineation of roles between the two branches.
>
> > Q12: In Section 3.2, a more detailed explanation of the definition of would be appreciated. Furthermore, the part around this section involving attributes is somewhat challenging to comprehend in written form. It might be a good idea to exclude this part if it ultimately relates to the explanation in Section 3.3.2.
> >
>
> A12: We have refined the manuscript in alignment with your feedback.
>
> > Q13: Are dx, dy, and dc in Section 3.3.1 uniquely determined for each node? Also, why is dc necessary?
> >
>
> A13: Thanks for the comment. dx, and dy are uniquely determined for each node as the node feature. We have added ablation study that removes the dx, dy to validate its effectiveness, which is shown as follows. It is obvious that this feature is helpful for our PDE solving.
>
> | Test set | 4-Corners L2 | 4-Corners MAE | 3-Corners L2 | 3-Corners MAE | 2-Corners L2 | 2-Corners MAE | 1-Corner L2 | 1-Corner MAE | No-Corner L2 | No-Corner MAE |
> | --- | --- | --- | --- | --- | --- | --- | --- | --- | --- | --- |
> | BENNO w/o dx,dy | 0.3654±0.1591 | 0.9989±0.3752 | 0.4535±0.2261 | 1.2912±0.5961 | 0.5549±0.2044 | 1.5689±0.5708 | 0.5797±0.2219 | 1.535±0.5039 | 0.5818±0.2219 | 1.7535±0.5547 |
> | BENO (ours) | 0.3523±0.1245 | 0.9650±0.3131 | 0.4308±0.1994 | 1.2206±0.4978 | 0.4910±0.1888 | 1.4388±0.5227 | 0.5416±0.2133 | 1.4529±0.4626 | 0.5542±0.1952 | 1.7481±0.5394 |

---

> ### Author Response · Authors · 2023-11-22
> **Response to Reviewer MqG8 (5)**
>
> > Q14: On page 5, is "B" ultimately equal to "B^1"? It would be beneficial to provide a more detailed explanation of where "t" changes and what its range is.
> >
>
> A14: **t** is the step idx ranging from 1 to 5 in our experiments. And B^t represents the boundary embedding in the **t-*th*** message passing step.
>
> > Q15: The definition of MAE for Table 2 is present in the appendix but not in the main text.
> >
>
> A15: Thanks for the catch. In the updated manuscript, we have added the definition of MAE for Table 2 to the main text.
>
> > Q16: In Section 4.3.2, the explanation of training at 32x32 and testing at 64x64, is this capability attributed to BENO's use of graph neural networks, or are there other features enabling this super-resolution? A more detailed explanation would be appreciated.
> >
>
> A16: This capability is primarily attributed to the GNN architecture used in BENO. The GNN's capability to handle super-resolution stems from its approximation of the Green's function. The Green's function describes how the value at one location influences the solution at the current position. This characteristic makes GNN particularly suitable for super-resolution tasks, as it adeptly models the interplay and impact of various spatial data points on each other.
>
> > Q17: In Appendix K, when looking at Figure 6, it appears that the results for "Ours" are different from other solution profiles, forming square blocks with discontinuities. What is the reason or cause behind this phenomenon?
> >
>
> A17: The key factor is the narrow range of numerical values in shown data. In such a scenario, even minor fluctuations in the predicted values near the corners can result in the values falling outside of a given scale interval on the color-bar. This creates the visual effect of discontinuous blocks, as opposed to a smoother gradient typically seen with a wider range of values.

---

> ### Author Response · Authors · 2023-11-23
> **A gentle reminder: please let us know if you have additional questions**
>
> Dear Reviewer MqG8,
>
> We appreciate your time to review and the constructive comments to encourage our work. We want to leave a gentle reminder due to nearing the end of the discussion period.
>
> We have tried to address all your concerns by providing more explanations and results. Please go over our response, and if you have additional questions, please let us know.
>
> Thank you,
>
> the Authors.

---

### Official Review · Reviewer_L19w · 2023-11-01

**Soundness:** 3 good
**Presentation:** 3 good
**Contribution:** 3 good
**Rating:** 6
**Confidence:** 4

**Summary:**

The paper presents a modified neural operator for solving elliptic PDEs where in the boundary conditions are independently encoded as part of the architecture. The basic idea relies on constructing two GNN based operators for the interior source and the boundary values respectively and further encoding the boundary geometry using a Transformer. The approach is tested on a dataset of geometries with homogenous and inhomogenous boundary conditions.

**Strengths:**

1. Overall the paper is well written and the details are clear and concise.

2. The method appears to generalize to a variety of boundary conditions that competing methods have trouble solving.

3, The experimental and ablation results are exhaustive and show the effect of each design choice.

**Weaknesses:**

1. The grid sizes for the domain are quite small (32x32, 64x64) when compared with FNO. I would like to see if the approach generalizes to larger and more practical grid sizes as well. This is especially important given the sequence length constraints that come with using transformers.

2. It would also be useful to get some additional information about training and inference times for BENO and baselines.

**Questions:**

See weaknesses above.

---

> ### Author Response · Authors · 2023-11-22
> **Response to Reviewer L19w**
>
> > Q1: The grid sizes for the domain are quite small (32x32, 64x64) when compared with FNO. I would like to see if the approach generalizes to larger and more practical grid sizes as well. This is especially important given the sequence length constraints that come with using transformers.
> >
>
> A1: Thanks for the comment. We have conducted additional super-resolution experiment, where the models are trained on 64x64 resolution, and are tested on irregular domains at a high **128x128 resolution**, pushing the boundaries of our BENO's applicability. The results are shown as below. Specifically, BENO consistently shows superior performance by a large margin.
>
> | Test set | 4-Corners L2 | 4-Corners MAE | 3-Corners L2 | 3-Corners MAE | 2-Corners L2 | 2-Corners MAE | 1-Corner L2 | 1-Corner MAE | No-Corner L2 | No-Corner MAE |
> | --- | --- | --- | --- | --- | --- | --- | --- | --- | --- | --- |
> | MP-PDE | 1.0543±0.2839 | 0.0997±0.0426 | 1.0475±0.2629 | 0.0982±0.0394 | 1.0206±0.1826 | 0.1291±0.0420 | 1.0144±0.1416 | 0.1645±0.0543 | 1.0063±0.1008 | 0.2257±0.0587 |
> | BENO (ours) | 0.5815±0.1276 | 0.0372±0.0236 | 0.6599±0.2923 | 0.0369±0.0197 | 0.6796±0.2039 | 0.0625±0.0280 | 0.6852±0.1869 | 0.0954±0.0452 | 0.7429±0.1594 | 0.1600±0.0508 |
>
> We see that our model still outperforms the MP-PDE baseline by a large margin, at this high resolution.
>
> > Q2: It would also be useful to get some additional information about training and inference times for BENO and baselines.
> >
>
> A2: Thank you for your comment. The average time taken for BENO and the graph-based baseline methods during a single epoch, computed over five iterations, is listed below. We can see that the modest increase in computational time of BENO is justified by the substantial performance improvements offered by the integration of the Transformer and dual-branch GNN structures. The added complexity of these components contributes to the enhanced capability of the BENO model, making it a feasible and efficient choice in scenarios where advanced performance is crucial.
>
> | Method | GNN-PDE | MP-PDE | BENO (ours) |
> | --- | --- | --- | --- |
> | Time | 0.0054 | 0.0092 | 0.0210 |

---

> > ### Comment · Reviewer_L19w · 2023-11-22
> > **Thanks for the response**
> >
> > I thank the authors for the response. The additional experiments on larger domains as well as the ones addressing other reviewers' comments are interesting and show that the proposed approach is promising. I also agree with the authors that the higher training time is adequately compensated with better performance. I appreciate the extra effort.

---

> > > ### Author Response · Authors · 2023-11-23
> > > **Response to Reviewer L19w**
> > >
> > > Thanks for your valuable feedback on our responses. We genuinely value your contributions and will ensure that any further suggestions will be carefully incorporated.

---

### Official Review · Reviewer_QC4z · 2023-11-04

**Soundness:** 3 good
**Presentation:** 3 good
**Contribution:** 3 good
**Rating:** 8
**Confidence:** 4

**Summary:**

This work introduces a novel neural operator architecture called BENO for solving elliptic PDEs with complex geometries and inhomogeneous boundary values. The BENO model outperforms state-of-the-art neural operators. The paper also discusses the potential impact of this research on various scientific and engineering domains, including computational fluid dynamics, solid mechanics, and electromagnetics.

**Strengths:**

1. The manuscript is well-written and easy to follow.
2. The paper introduces a novel neural operator architecture that embeds complex geometries and inhomogeneous boundary values into the solving of elliptic PDEs.
3. The BENO model outperforms state-of-the-art neural operators and strong baselines.

**Weaknesses:**

1. Lack of comparison with more SOTA works like multiwavelet neural operator.
2. Focusing on solving elliptic PDEs with complex geometries and inhomogeneous boundary values may limit the applications.

**Questions:**

1. How can the dataset used to evaluate the BENO model be expanded to include more diverse scenarios and improve the generalization performance of the model?
2. Would the authors be able to provide the number of parameters to quantify each model's complexity?
3. How can the BENO model be adapted to handle time-dependent PDEs or other types of dynamic systems?
4. Can the BENO model be used to generate physically meaningful insights and interpretability?
5. There are some latest related operator works that should be referred to for a boarder audience:[Xiao, Xiongye, et al. "Coupled Multiwavelet Neural Operator Learning for Coupled Partial Differential Equations."] [Gupta, Gaurav, et al. "Multiwavelet-based operator learning for differential equations."]; [Gupta, Gaurav, et al. "Non-linear operator approximations for initial value problems."].

---

> ### Author Response · Authors · 2023-11-22
> **Response to Reviewer QC4z (1)**
>
> > Q1: Lack of comparison with more SOTA works like multiwavelet neural operator.
> >
>
> A1: Thanks for your insightful feedback. We have added the ****Multiwavelet-based Operator**** as the baseline, and implement experiments under our main setting, which is trained on 900 4-corners samples and tested on 5 datasets. The results are shown as below and it is obvious that our BENO surpass the baseline by a large margin.
>
> | Test set | 4-Corners L2 | 4-Corners MAE | 3-Corners L2 | 3-Corners MAE | 2-Corners L2 | 2-Corners MAE | 1-Corner L2 | 1-Corner MAE | No-Corner L2 | No-Corner MAE |
> | --- | --- | --- | --- | --- | --- | --- | --- | --- | --- | --- |
> | MWTNO | 1.0264±0.2678 | 2.5639±0.3871 | 1.0465±0.2741 | 2.7752±0.3216 | 1.0433±0.3301 | 2.7399±0.3186 | 1.0428±0.2973 | 2.7350±0.3119 | 1.0397±0.2711 | 2.7015±0.3228 |
> | BENO (ours) | 0.3523±0.1245 | 0.9650±0.3131 | 0.4308±0.1994 | 1.2206±0.4978 | 0.4910±0.1888 | 1.4388±0.5227 | 0.5416±0.2133 | 1.4529±0.4626 | 0.5542±0.1952 | 1.7481±0.5394 |
>
> > Q2: Focusing on solving elliptic PDEs with complex geometries and inhomogeneous boundary values may limit the applications.
> >
>
> A2:  Despite the focus on elliptic PDEs, we believe this is an important and critical research that can be applicable to a wide range of natural phenomena and engineering applications.
>
> - **Importance of boundary conditions for any PDE systems.** It is known that any PDEs can be characterized as parabolic, hyperbolic, and elliptic PDEs, which further can be categorized as initial-boundary value problem (IBVP), and initial value problem (IVP) and boundary value problem (BVP), respectively. For any of these PDEs, the treatment of boundary conditions plays a critical role in accurately modeling natural phenomena and engineering applications. While we focus on the application of BENO on elliptic PDEs, we believe that the boundary condition treatment and incorporation into the overall solver is applicable to other PDE systems.
> - **Applicability of elliptic PDEs to the real-world phenomena.** The utility of elliptic PDEs is not limited to steady problems, e.g., we can obtain unsteady solution of nonlinear PDEs by repeated solution of elliptic PDEs. For instance, the pressure-Poisson equation (which is an elliptic PDE) for incompressible flows can be applied to unsteady phenomena. Various plasma simulations uses Poisson solver that is time dependent. Helmholtz equation is used in a variety of phenomena, including electromagnetic radiation, seismology, and acoustics. These systems can have complex geometries, e.g., airfoil, propulsion devices, plasma sources for semiconductor manufacturing, with different types of materials (e.g., dielectric, conducting, catalytic, chemically reactive) and inhomogeneous boundary values (e.g., dependent on space and time). Thus, we believe that demonstration of BENO on elliptic PDEs has immense impact to a large number of applications. This introduction is updated to emphasize this point. Besides, we have revised the manuscript to delete the statements that elliptic PDEs are only applicable for steady-state phenomena.
>
> Furthermore, to validate the generalization ability of our BENO, we **also supplement more experiments** on other scenarios in the following part.

---

> ### Author Response · Authors · 2023-11-22
> **Response to Reviewer QC4z (2)**
>
> > Q3: How can the dataset used to evaluate the BENO model be expanded to include more diverse scenarios and improve the generalization performance of the model?
> >
>
> A3: Thanks for your comment. We have generated new datasets on different boundary condition (i.e., **Neumann BCs**) type and different types of solution operators (i.e., **Darcy flow**) under our existing setting with different corner cases. Typically boundary conditions can be categorized into Dirichlet and Neumann BCs, or the combination of the two. We believe that the addition of Neumann BCs to the manuscript makes the paper to be applicable to a much wider range of systems. We represent the supplementary experiment results as follows:
>
> - Performances of our BENO and the compared baseline under **Neumann boundary condition** are shown below, which are **trained on 900 4-corners** samples and tested on 5 datasets under relative L2 norm and MAE separately. The whole comparison table with all the baselines is shown in **Appendix L**. We see that our method significantly outperforms the MP-PDE baseline in this setting.
>
> | Test set | 4-Corners L2 | 4-Corners MAE | 3-Corners L2 | 3-Corners MAE | 2-Corners L2 | 2-Corners MAE | 1-Corner L2 | 1-Corner MAE | No-Corner L2 | No-Corner MAE |
> | --- | --- | --- | --- | --- | --- | --- | --- | --- | --- | --- |
> | MP-PDE | 1.0070±0.0813 | 2.3595±0.6941 | 0.9895±0.0973 | 2.5480±0.8955 | 1.0134±0.1120 | 2.7345±0.8393 | 0.9782±0.1240 | 2.9679±0.7958 | 0.9670±0.1164 | 3.4807±1.1143 |
> | BENO (ours) | 0.3568±0.0988 | 0.8311±0.2864 | 0.4201±0.1170 | 1.0814±0.3938 | 0.5020±0.1648 | 1.3918±0.5454 | 0.5074±0.1422 | 1.5676±0.4815 | 0.5221±0.1474 | 1.8649±0.5472 |
> - Performances of our BENO and the compared baseline under **Neumann boundary condition** are shown below, which are **trained on 900 mixed** samples (180 samples each from 5 datasets) and tested on 5 datasets under relative L2 norm and MAE separately. The whole comparison table with all the baselines is shown in **Appendix L**. Again, we see that our method significantly outperforms the MP-PDE baseline in this setting
>
> | Test set | 4-Corners L2 | 4-Corners MAE | 3-Corners L2 | 3-Corners MAE | 2-Corners L2 | 2-Corners MAE | 1-Corner L2 | 1-Corner MAE | No-Corner L2 | No-Corner MAE |
> | --- | --- | --- | --- | --- | --- | --- | --- | --- | --- | --- |
> | MP-PDE | 1.0245±0.1048 | 2.3973±0.7015 | 0.9989±0.1277 | 2.5510±0.8717 | 1.0277±0.1399 | 2.7722±0.8091 | 0.9940±0.1543 | 2.9998±0.7781 | 0.9731±0.1414 | 3.4930±1.0867 |
> | BENO (ours) | 0.4237±0.1237 | 1.0114±0.4165 | 0.3970±0.1277 | 1.0378±0.4221 | 0.3931±0.1347 | 1.0881±0.3993 | 0.3387±0.1279 | 1.0520±0.4253 | 0.3344±0.1171 | 1.2261±0.4467 |
> - Performances of our BENO and the compared baseline on **Darcy flow** are shown below**,** which are **trained on 900 4-corner** samples and tested on 5 datasets under relative L2 norm and MAE separately. The whole comparison table with all the baselines is shown in **Appendix M**.
>
> | Test set | 4-Corners L2 | 4-Corners MAE | 3-Corners L2 | 3-Corners MAE | 2-Corners L2 | 2-Corners MAE | 1-Corner L2 | 1-Corner MAE | No-Corner L2 | No-Corner MAE |
> | --- | --- | --- | --- | --- | --- | --- | --- | --- | --- | --- |
> | MP-PDE | 0.4802±0.1840 | 0.3269±0.2085 | 0.5332±0.1742 | 0.4652±0.2999 | 0.6197±0.1709 | 0.6307±0.3282 | 0.6906±0.1432 | 0.8469±0.4087 | 0.7406±0.1271 | 1.0906±0.3949 |
> | BENO (ours) | 0.2431±0.0895 | 0.1664±0.0773 | 0.2542±0.1252 | 0.2150±0.1270 | 0.2672±0.1497 | 0.2585±0.1313 | 0.2466±0.1405 | 0.3091±0.2350 | 0.2366±0.1104 | 0.3591±0.2116 |
>
> > Q4: Would the authors be able to provide the number of parameters to quantify each model's complexity?
> >
>
> A4: We would like to present the parameter count for our BENO model. Specifically, the Transformer component, dedicated to encoding the boundary, comprises 0.8M parameters, while each branch of the GNN-based network encompasses 1.14M parameters. This indicates that the complexity of our approach is entirely manageable and within an acceptable range.
>
> | Method | GKN | FNO | GNN-PDE | MP-PDE | BENO(ours) |
> | --- | --- | --- | --- | --- | --- |
> | Parameters | 1.09M | 0.44M | 0.84M | 0.63M | 3.08M |
>
> > Q5: How can the BENO model be adapted to handle time-dependent PDEs or other types of dynamic systems?
> >
>
> A5: As we responded in A2, even some elliptic PDEs are designed to handle time-dependent solutions and the importance of boundary condition treatment remains same for other dynamic systems. We have modified our introduction to highlight this point.

---

> ### Author Response · Authors · 2023-11-22
> **Response to Reviewer QC4z (3)**
>
> > Q6: Can the BENO model be used to generate physically meaningful insights and interpretability?
> >
>
> A6: The innovativeness of BENO is in accounting for the boundary condition treatments in the PDE solutions obtained from boundary-embedded GNN. Handling of the boundary conditions is a requirement for any PDE equations. Elliptic PDEs are good test problems to test such boundary condition effects (cf. boundary value problem). In addition, without accurately capturing the boundary conditions, the solutions for the PDEs obtained from a solver are not physically meaningful and cannot be applied to real systems. Hence, we believe that BENO’s strengths are in producing an acceptable tolerance with much faster computational speed, which generate physically meaningful results and enhance the interpretability of the solutions.
>
> > Q7: There are some latest related operator works that should be referred to for a boarder audience:
> >
>
> A7: We have added [Xiao, Xiongye, et al. "Coupled Multiwavelet Neural Operator Learning for Coupled Partial Differential Equations."], [Gupta, Gaurav, et al. "Multiwavelet-based operator learning for differential equations."], and [Gupta, Gaurav, et al. "Non-linear operator approximations for initial value problems."] in the revised version of **Related Work** for a boarder audience.

---

> > ### Comment · Reviewer_QC4z · 2023-11-22
> > **I will stick to the acceptance recommendation.**
> >
> > I appreciate the detailed response with comprehensive results compared with MWT. Since the authors have addressed most of my concerns, I will raise my score and maintain my recommendation for acceptance.

---

> > > ### Author Response · Authors · 2023-11-22
> > > **Response to Reviewer QC4z**
> > >
> > > Thanks for your valuable feedback on our paper and raising of score. We genuinely value your contributions and will ensure that any further suggestions will be carefully incorporated.

---

### Author Response · Authors · 2023-11-22
**General Response**

We thank the reviewers for the thorough reviews and constructive suggestions. We acknowledge the positive comments such as “**well-written”** (Reviewer QC4z, L19w, tbt5, 113v), “**a novel neural operator”** (reviewer Qc4z), “**outperforms the benchmarks”** (Reviewer QC4z, tbt5), “**exhaustive experiments and ablation”** (Reviewer L19w), and “**significance in handling complex problems”** (Reviewer MqG8, tbt5, 113v). We also believe that our **BENO** would significantly contribute to the community.

We have thoroughly revised our article in light of the insightful feedback received, and we provide a brief summary of these significant modifications below. All major changes have been prominently highlighted in blue to facilitate easy identification and review.

- We have enriched our research by integrating **two new baselines**: DeepONet and a Multiwavelet-based operator, applied to our datasets for a more comprehensive analysis. We see that our BENO outperforms these two strong baselines, which demonstrates the effectiveness of BENO. For more details, see the responses to Reviewer QC4z and MqG8.
- Our study now includes experiments under **Neumann boundary conditions**. These experiments involve training on both 4-corners and mixed samples, followed by testing across five distinct datasets. Our BENO surpasses all baseline models, showcasing its superior performance and generalization capabilities. For additional information, please refer to the responses to Reviewer QC4z, MqG8, and tbt5.
- We have expanded our scope to encompass experiments on **Darcy flow**. In our experiments, our BENO proficiently maps coefficients to solutions. The empirical outcomes corroborate the model's exceptional versatility, significantly outperforming the foremost baseline. For an in-depth examination, please see responses provided to Reviewer QC4z, MqG8, and tbt5.
- To assess the generalization capabilities of our model, we have conducted tests on irregular domains at a high **128x128 resolution**, pushing the boundaries of our BENO's applicability. We can see that our BENO outperforms the baseline by a wide margin. Please see response to Reviewer L19w for details.
- We have undertaken a comprehensive **improvement on the baseline** to comprehensively explore its potential. This ensures a fair and equitable comparison by fully harnessing the capabilities of the baseline.
- An ablation study has been conducted to analyze the effects of removing **dx and dy** components, providing deeper insights into the model's sensitivity to validate the effectiveness.
- We have explored variations in our model architecture through an **ablation on the choice of Transformer**. This involved utilizing two GNN branches to connect the boundary and interior, serving as an alternative to the Transformer and broadening our understanding of different architectural impacts.
- We have enriched our analysis by **visualizing the outputs from two distinct branches**, which has significantly improved our comprehension of our architecture's functionality.

---

### Meta-Review · Area_Chair_odWX · 2023-12-06

**Metareview:**

The paper proposes a new neural operator approach for solving elliptic PDEs. The main contribution is to alleviate the challenge of complex geometries and inhomogeneous boundary values using a combination of an auxiliary GNN and a transformer encoder. Overall, the authors do a very good job in justifying their approach in terms of comparisons with existing baselines.

The paper received a somewhat-high initial spread in review scores. However, the authors were able to address most of the concerns raised by the reviewers, and as a result of internal discussions, the overall sentiment among the reviewers (and myself) seems generally quite supportive of acceptance. For the final revision, the authors should consider buttressing their paper with additional results showing performance on a broader range of shapes and more intricate domains.

**Justification For Why Not Higher Score:**

The main technical innovation seems to be carefully piecing together various ideas from the literature, and as such seems better suited as a poster.

**Justification For Why Not Lower Score:**

Contribution to the field of neural PDE solvers seems quite clear and above the bar for acceptance.

---

### Decision · Program_Chairs · 2024-01-16

Accept (poster)